# Revealing β-TrCP activity dynamics in live cells with a genetically encoded biosensor

Debasish Paul [1], Stephen C. Kales[2], James A. Cornwell[1], Marwa M. Afifi [1], Ganesha Rai[2], Alexey Zakharov[2], Anton Simeonov[2] & Steven D. Cappell [1] ✉

The F-box protein beta-transducin repeat containing protein (β-TrCP) acts as a substrate adapter for the SCF E3 ubiquitin ligase complex, plays a crucial role in cell physiology, and is often deregulated in many types of cancers. Here, we develop a fluorescent biosensor to quantitatively measure β-TrCP activity in live, single cells in real-time. We find β-TrCP remains constitutively active throughout the cell cycle and functions to maintain discreet steady-state levels of its substrates. We find no correlation between expression levels of β-TrCP and β-TrCP activity, indicating post-transcriptional regulation. A high throughput screen of small-molecules using our reporter identifies receptor-tyrosine kinase signaling as a key axis for regulating β-TrCP activity by inhibiting binding between β-TrCP and the core SCF complex. Our study introduces a method to monitor β-TrCP activity in live cells and identifies a key signaling network that regulates β-TrCP activity throughout the cell cycle.

Ubiquitination, which is governed by the ubiquitin-proteasome system (UPS), is one of the most prevalent post-translational modifications and regulates various cellular processes such as cellular signaling, cell cycle progression, metabolism, transcription, translation, and cell death[1–3]. Ubiquitination of proteins is carried out via three major steps through the activity of three classes of enzymes. In the first step, ubiquitin is activated by a ubiquitin-activating enzyme E1 in the presence of ATP. In the second step, the activated ubiquitin is transferred to the ubiquitin-conjugating enzyme E2, and in the third step, the activated ubiquitin is transferred to a substrate through the ubiquitin ligase enzyme E3[4]. Critically, E3 ubiquitin ligases impart the substrate specificity for ubiquitination and subsequent downstream signaling or protein degradation[4]. There are more than 600 ubiquitin ligases encoded by the human genome (representing roughly 5% of human genes), underscoring their importance in many biological processes[5]. One of the largest families of ubiquitin ligases are the multi-protein SCF (**S**kp1, **C**ullin, **F**-box) ubiquitin ligases, which have been shown to target multiple oncogenes and tumor suppressors for ubiquitin-mediated degradation[6,7]. Within the SCF protein complex, the F-box protein functions as the substrate specifying subunit and is

responsible for directing the ubiquitination of numerous proteins essential for cellular function[7]. Due to their ability to regulate the expression and activity of oncogenes and tumor suppressor genes, F-box proteins play important roles in development and progression of multiple diseases including cancer[6].

The SCF in complex with the F-box protein β-transducin repeat-containing protein (β-TrCP) is a well-characterized E3 ubiquitin ligase that recognizes a wide variety of substrates[8–10]. A majority of the canonical substrates contain the consensus sequence DpSGX$_{1–3}$pS (X represents any amino acid) or its variants, and the phosphorylation of serine residues by a specific kinase is required for substrate recognition by β-TrCP and subsequent ubiquitination[6,11,12]. Non-canonical substrates contain the phospho-mimetic consensus sequence DDGXXD and do not require prior phosphorylation for substrate recognition by β-TrCP[13]. It has been demonstrated that β-TrCP has two homologs, β-TrCP1 and β-TrCP2, which share near identical biological functions and are believed to be redundant with respect to substrate regulation[14,15].

β-TrCP plays vital roles in cell cycle control, cell growth and apoptosis, angiogenesis, DNA damage repair, metabolism, and NF-kB

[1]Laboratory of Cancer Biology and Genetics, Center for Cancer Research, National Cancer Institute, Bethesda, MD 20892, USA. [2]National Center for Advancing Translational Sciences, National Institutes of Health, Rockville, MD 20850, USA. ✉e-mail: steven.cappell@nih.gov

signaling by recognizing specific substrates such as Wee1, Emi1, Cdc25, BimEL, VEGFR2, Set8, and IkB[16–21]. In addition, mis-regulation of β-TrCP is associated with cancer initiation and progression[22]. Elevated β-TrCP protein and mRNA expression are verified in 56% of colorectal cancer tissues and are correlated with poor clinical prognosis[23]. Interestingly, knocking down β-TrCP1,2 in combination with chemotherapeutic drugs can improve the efficacy of killing cancer cells[24], which makes β-TrCP an attractive target for cancer treatment[25]. Despite the important role β-TrCP plays in both normal and cancer physiology, relatively little is known about how β-TrCP activity is regulated in cells. Canonical substrates must first be phosphorylated prior to being ubiquitinated by β-TrCP, which means most studies focus on the regulation of the relevant kinases, rather than on β-TrCP itself. However, there is regulation of the SCF complexes as well, which can influence the extent to which substrates are degraded. For example, SCF complexes are regulated at the level of complex formation via the protein CAND1 and via NEDD8 conjugation of the Cullin subunit[26–29]. In order to investigate the regulation of β-TrCP and other E3 ligases in both normal and cancer cells, new tools are needed.

Here, we develop a fluorescent biosensor capable of reporting the real-time activity of β-TrCP in live, single cells. This reporter shows selective specificity and sensitivity for β-TrCP compared to other similar SCF complexes. Using time-lapse microscopy, we find that while β-TrCP activity is relatively constant throughout different phases of the cell cycle (G1, S, G2, and mitosis), β-TrCP activity is highly elevated during quiescence or G0 phase. Upon mitogen stimulation and subsequent re-entry into the cell cycle, β-TrCP activity is rapidly inhibited due to decreased association of β-TrCP with the core SCF complex as opposed to increased β-TrCP protein levels. Consistent with this observation, we find no significant correlation between the activity of β-TrCP and levels of β-TrCP protein in single cells within a population as well as across various non-cancerous and cancerous cell lines, indicating that β-TrCP activity is primarily regulated at the level of complex formation with the SCF and not by the relative levels of β-TrCP protein. Importantly, we find β-TrCP functions primarily to control the steady-state levels of its substrates, rather than degrade them completely like other ubiquitin ligases. We construct a luciferase version of the β-TrCP reporter and conduct a high-throughput screen of small-molecule inhibitors, identifying receptor tyrosine kinase signaling as a key modulator of β-TrCP activity that can tune the steady-state levels of its substrates. In the future, understanding the dynamics of β-TrCP activity will lead to new insights into its biological function and help lead to the development of effective β-TrCP inhibitors and activators.

## Results

### Construction of a generalizable β-TrCP reporter

Due to its importance in both normal and pathological cell physiology, β-TrCP has previously been well characterized biochemically, and many substrates have been identified (Fig. 1a)[13]. While most studies involving β-TrCP focus on the dynamic regulation of its substrates, little is known about how the activity of β-TrCP itself is dynamically regulated in cells. Therefore, we sought to design a genetically encoded fluorescent reporter that would allow us to measure real-time β-TrCP activity in live, single cells. To construct the β-TrCP reporter, we employed a strategy similar to the one first reported by Sawkaue-Sawano et. al. and utilized by others, which involves fusing an optimal canonical degron motif to a fluorescent protein[30–32]. For β-TrCP, the canonical degron motif is DpSGXXpS, where the phosphorylated serine residues are recognized by β-TrCP (Fig. 1b)[33]. While many β-TrCP substrates contain this common motif, different kinases phosphorylate each substrate. Thus, any reporter designed using this canonical degron motif will have the limitation of being a co-incidence detector for both β-TrCP activity as well as the activity of another kinase, preventing the direct measurement of β-TrCP activity and complicating

the interpretation of any results. Fortunately, in addition to the canonical degron motif, there are also β-TrCP substrates that contain a non-canonical degron-motif DDGxxD (eg; CDC25B)[18], which contain aspartic acid residues in place of the phosphorylated serines. The negatively charged aspartic acid residues presumably mimic the negative charge imparted by the addition of a phosphate group on serine. Thus, substrates containing the non-canonical degron are constitutively recognized and targeted by β-TrCP and could serve as the basis for a universal β-TrCP activity reporter.

Having chosen to focus our efforts on substrates containing non-canonical degrons, we fused the fluorescent protein mVenus to various fragments of human CDC25B, which contains a non-canonical β-TrCP degron motif (Fig. 1c and Supplementary Fig. 1a). All constructs were made in a lentivirus backbone and under the control of a constitutive EF1α promoter. Since human CDC25B is also degraded by the ubiquitin ligase anaphase-promoting complex/cyclosome (APC/C) (Supplementary Fig. 1a)[34], we chose a region of CDC25B that only contained the non-canonical degron (DDGFVD, aa268–273) and lacked the APC/C degron (KEN, aa192–195) or the C-terminal catalytic domain (aa356–580). To determine the shortest possible degron region capable of being degraded by β-TrCP, we made several constructs using variable length fragments of CDC25B. We added a nuclear localization sequence (NLS) to the N-terminus of some of the constructs because the endogenous NLS of CDC25B was not located within those fragments[35]. Constructs were scored based on several parameters: (i) whether the fluorescence signal increased after β-TrCP knockdown or treatment with the pan-SCF inhibitor MLN-4924[36], (ii) the dynamic range of this change, (iii) sub-cellular localization, and (iv) the impact of construct expression on normal cell cycle[33] (Fig. 1c and Supplementary Fig. 1b). We found that a construct containing an NLS fused to CDC25B aa198–338 (hereafter referred to as β-TRCP reporter) showed the highest dynamic range when we knocked down β-TrCP1,2 using siRNA, and had no effect on the normal cell cycle or intermitotic time (Supplementary Fig. 1c, d). Additionally, we did not observe any change in β-TrCP expression or activity when comparing parental cells to cells expressing the β-TrCP reporter (Supplementary Fig. 1e). Notably, the addition of the NLS generally improved the dynamic range of the reporter, (Fig. 1c and Supplementary Fig. 1b) without disrupting the localization of the reporter in both the nucleus and the cytoplasm (Supplementary Fig. 1f).

We next sought to validate our β-TrCP reporter using several methods to perturb either β-TrCP expression or activity. We quantified the β-TrCP reporter levels in hundreds of HeLa cells using live-cell microscopy and quantitative image analysis. Knocking down β-TrCP1,2 using siRNA caused a large increase in the fluorescence signal of the reporter, consistent with reduced β-TrCP-mediated degradation (Fig. 1d, e). Conversely, overexpressing β-TrCP1 but not a β-TrCP1 mutant lacking an F-box domain (ΔF-β-TrCP) resulted in a reduction in the fluorescence signal of the reporter, consistent with increased β-TrCP-mediated degradation. Treatment of cells with MLN-4924 also resulted in an increase in the reporter levels (Fig. 1f, g). Importantly, treatment with cycloheximide (protein translation inhibitor) resulted in almost a complete loss of reporter levels indicating that our β-TrCP reporter is indeed constitutively degraded. Furthermore, we measured the synthesis rate of the reporter throughout the cell cycle and found it was synthesized at a constant rate, independent of cell cycle phase (Supplementary Fig. 1g). Taken together, these results argue that changes in the β-TrCP reporter concentration within single cells are a direct reflection of β-TrCP activity.

### β-TrCP specifically binds to and ubiquitinates the β-TrCP reporter

To further validate our β-TrCP reporter, we tested the specificity and sensitivity of the developed reporter towards β-TrCP in vitro. We performed co-immunoprecipitation experiments and found that

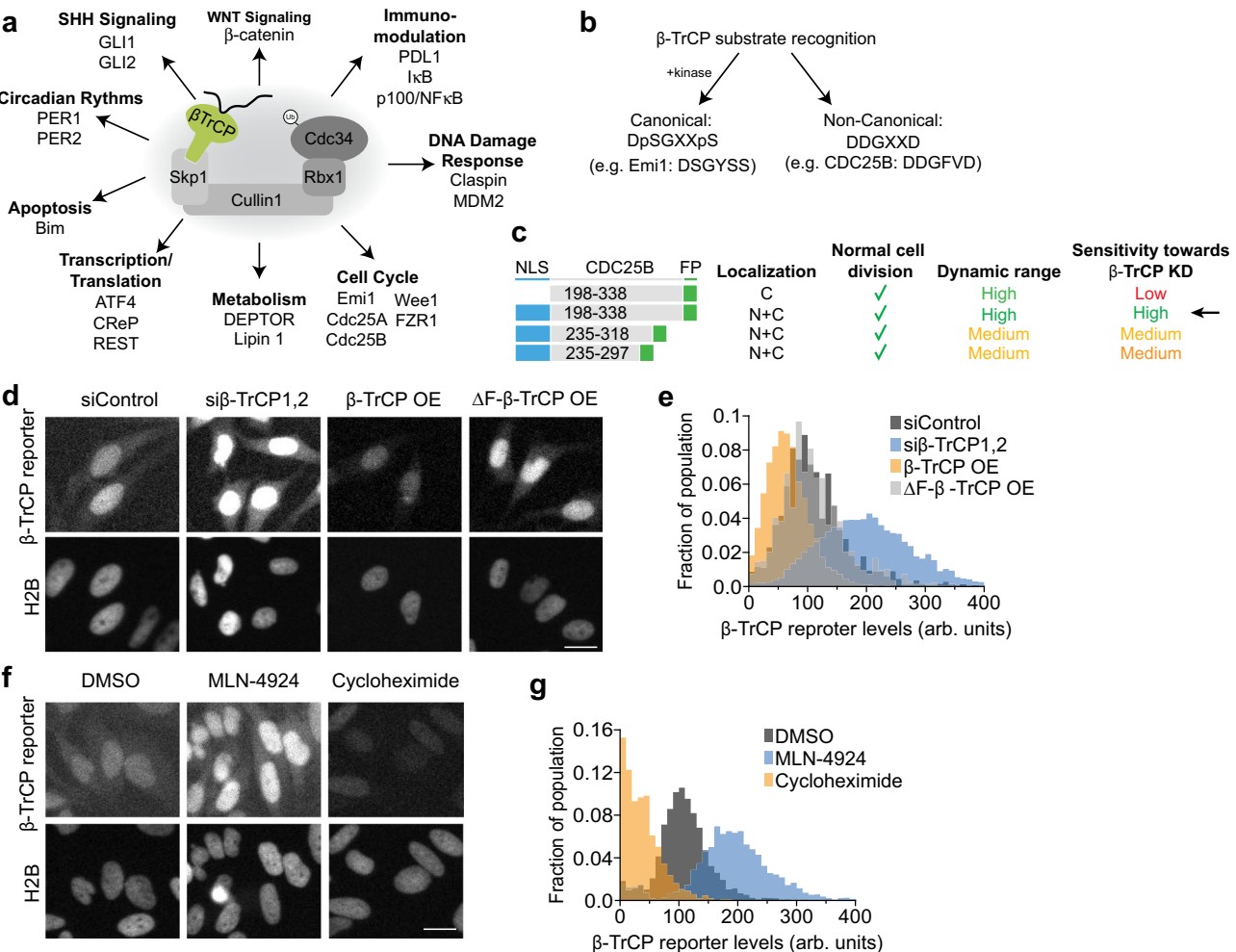

**Fig. 1 | Construction of a generalizable β-TrCP reporter. a** Schematic showing several known substrates of β-TrCP and the pathways it regulates. **b** β-TrCP recognizes two types of degron sequences. Canonical substrates like Emi1 contain the degron DpSGXXpS whereby the S or serine residue must be phosphorylated to bind to β-TrCP. Non-canonical substrates like CDC25B contain a degron consisting of phospho-mimetic amino acids (D or E) in place of serine residues. For example, CDC25B contains a DDGFVD degron sequence. **c** Various constructs with fragments of human CDC25B fused to mVenus. Each construct was evaluated using the criteria listed. The construct with the best attributes was NLS-CDC25B (aa198–338)-mVenus (hereafter β-TrCP reporter). NLS nuclear localization signal, C cytoplasmic, N nuclear, √ = Pass, x = Fail. **d** HeLa cells expressing the β-TrCP reporter and H2B-mTurquoise to visualize the nucleus. Cells were treated as indicated with either β-TrCP1,2 siRNA, transfected with full-length β-TrCP1 (β-TrCP OE) or β-TrCP with the F-box deleted (ΔF-β-TrCP1 OE). Images were taken 48 h after transfection. Scale bar is 10 μm. Representative images from *n* = 3 experiments. **e** Histograms depicting β-TrCP reporter levels in single cells from (**d**) *N* = 2431 (siControl), 18,506 (siβ-TrCP), 16,557 (β-TrCP OE), and 1380 (ΔF-β-TrCP1 OE) cells. **f** HeLa cells expressing the β-TrCP reporter and H2B-mTurquoise to visualize the nucleus. Cells were treated with MLN-4924 (3 μM), cycloheximide (100 μg/mL), NCS (200 ng/mL), or DMSO for 4 h before capturing the image. Scale bar is 10 μm. Representative images from *n* = 3 experiments. **g** Histograms depicting β-TrCP reporter levels in single cells from (**f**) *N* = 8180 (DMSO), 6749 (MLN-4924), 9118 (Cycloheximide) cells. Source data for all figure panels are provided as a Source Data file.

β-TrCP pulled down with the β-TrCP reporter but not with a mutated version of the β-TrCP reporter where the degron sequence that is recognized by β-TrCP was mutated (DDGFVD mutated to AAGFVA; hereafter mutant β-TrCP reporter) (Fig. 2a and Supplementary Fig. 2a). Our β-TrCP reporter also failed to pull-down the similar F-box proteins FBXW5, FBXW7, FBXO25, and FBXO31, demonstrating that the reporter indeed specifically binds only to β-TrCP1 and 2 (Fig. 2a). Overexpressing β-TrCP led to a decrease in reporter levels, and this decrease could be rescued by treatment with MLN-4924 (Fig. 2b and Supplementary Fig. 2b, c). Furthermore, cells expressing a mutant β-TrCP lacking its F-box domain did not show a similar reduction in β-TrCP reporter levels as cells expressing wild-type β-TrCP (Fig. 2c). These data demonstrate that the degradation of the β-TrCP reporter is β-TrCP-mediated and SCF-dependent.

Next, we assessed whether β-TrCP promotes the polyubiquitination of the β-TrCP reporter. We co-transfected HEK 293T cells with β-TrCP and either the β-TrCP reporter or the mutant β-TrCP reporter.

After pulling down on either endogenous ubiquitin or His-tagged ubiquitin, we observed high-molecular weight ubiquitin conjugates on the wild-type but not mutant β-TrCP reporter (Fig. 2d and Supplementary Fig. 2d). Similarly, we found that overexpressing β-TrCP but not a mutant β-TrCP lacking its F-box domain resulted in high-molecular-weight polyubiquitin conjugates on the β-TrCP reporter (Fig. 2e). Importantly, we also observed high-molecular weight conjugates on the β-TrCP reporter in cells stably expressing endogenous β-TrCP (Fig. 2f, lane 1). As a further confirmation that endogenous β-TrCP mediates the polyubiquitination of the β-TrCP reporter, we observed a reduction in high-molecular-weight polyubiquitin conjugates on the β-TrCP reporter when we treated cells with either β-TRCP1,2 siRNA or Cullin1 (a core SCF component) siRNA (Fig. 2f, lanes 2 and 3). Finally, we obtained similar results when immunoprecipitating either the β-TrCP reporter or the mutant β-TrCP reporter and immunoblotting for ubiquitin (Supplementary Fig. 2e–g). Taken together, these data demonstrate that the β-TrCP reporter binds to

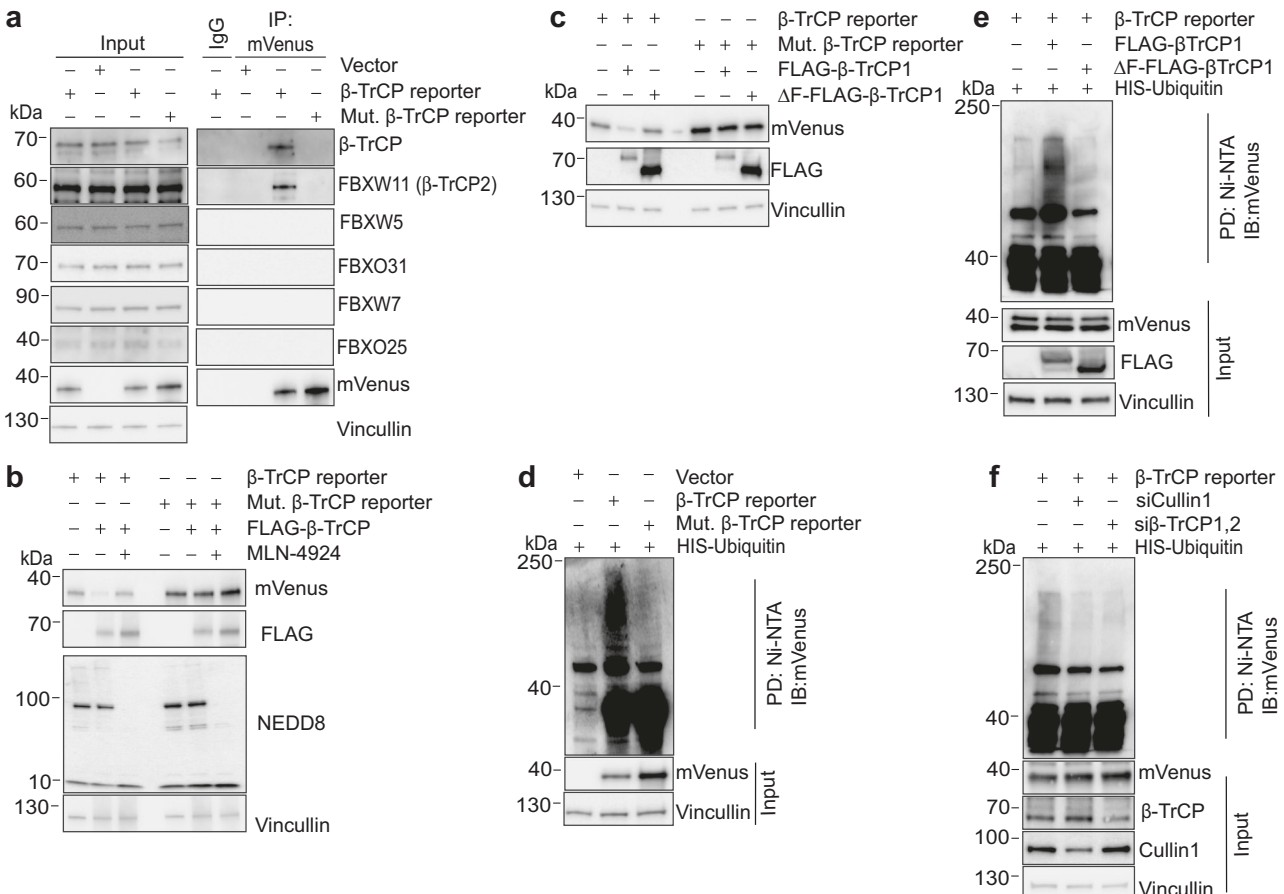

**Fig. 2 | β-TrCP specifically binds to and ubiquitinates the β-TrCP reporter.**
**a** HEK-293T cells were transfected as indicated with empty vector, β-TrCP reporter, or mutant β-TrCP reporter. Transfected cells were treated with 3 μM MLN-4924 for the 6 h before collection. Whole-cell extracts were immunoprecipitated with the indicated antibody followed by immunoblotting with the indicated antibody. Representative blot from *n* = 3 experiments. **b** HEK-293T cells were transfected as indicated. Cells were treated either with DMSO or MLN-4924 (3 μM) for 6 h before collection. Cells were harvested and whole-cell protein extracts were immuno-blotted for indicated antibodies. Representative blot from *n* = 3 experiments. **c** HEK-293T cells were transfected as indicated. Cells were harvested 48 h after transfection and whole-cell protein extracts were immunoblotted for the indicated antibodies. Representative blot from *n* = 3 experiments. **d** HEK-293T cells were transfected as indicated. Transfected cells were then treated with MG132 (5 μM) for

6 h. Whole-cell protein extracts were immunoprecipitated with Ni-NTA resins and immunoprecipitates were immunoblotted for mVenus. High mass ladder indicates polyubiquitination. Representative blot from *n* = 3 experiments. **e** HEK-293T cells were transfected as indicated. Transfected cells were then treated with MG132 (5 μM) for 6 h. and then whole-cell lysate were precipitated with Ni-NTA resins. Precipitates were probed for mVenus antibody. Representative blot from *n* = 3 experiments. **f** HeLa cells stably expressing the mVenus-β-TrCP reporter, were transfected with the indicated siRNAs against β-TrCP1,2 or Cullin1. Cells were treated with MG132 (5 μM) for the 6 h before collection. Whole-cell protein extracts were precipitated with Ni-NTA resins and immunoblotted for mVenus antibody. Representative blot from *n* = 2 experiments. Source data for all figure panels are provided as a Source Data file.

β-TrCP and is polyubiquitinated by β-TrCP in cells, leading to its degradation.

## β-TrCP activity is dynamically regulated during cell cycle entry and exit

Having validated that the β-TrCP reporter is regulated specifically by β-TrCP using both fixed-cell and biochemical assays, we next investigated how the β-TrCP reporter is dynamically regulated in live cells. We performed live-cell, time-lapse imaging of HeLa cells as they asynchronously went through the cell cycle, and we used an automated image analysis pipeline to quantify the reporter fluorescence in the nucleus. We compared the β-TrCP reporter dynamics to two previously characterized and validated ubiquitin ligase reporters for APC/C and SCF[Skp2] [30]. Both the reporters for APC/C and SCF[Skp2] were completely degraded in G1 and G2 phase respectively, when these ubiquitin ligases are known to be fully active (Supplementary Fig. 3a–c), consistent with previously published results[30,37]. Strikingly, we found that unlike these other E3 ubiquitin ligase-based reporters[32], levels of the β-TrCP reporter did not significantly change throughout the cell cycle,

but instead, remained at a relatively intermediate steady-state level (Fig. 3a–c). There was a slight increase in the median reporter levels during S/G2 phase, suggesting that β-TrCP activity may be moderately reduced in S/G2 phase compared to G1 phase (Fig. 3c). Notably, we observed a rapid spike in the median nuclear fluorescence intensity during mitosis, but this is likely due to a well-known artifact whereby mitotic cells round up and the fluorescent protein is concentrated in a smaller volume giving the false impression that the fluorescence intensity has suddenly increased[38,39]. Consistent with our previous results, we found that treating cells with β-TrCP1,2 siRNA or the SCF component Cullin1 siRNA resulted in higher steady-state levels of the β-TrCP reporter (Fig. 3d and Supplementary Fig. 3d–g). Treating cells with FBXW7 siRNA had no effect on the β-TrCP reporter dynamics, re-confirming that our reporter is specific to β-TrCP activity. Together, these data indicate that β-TrCP activity functions to maintain its substrates at discreet steady-state levels rather than completely degrading them.

Previous studies have shown that β-TrCP1 and 2 are redundant towards multiple substrates but that β-TrCP2 is the dominant paralog

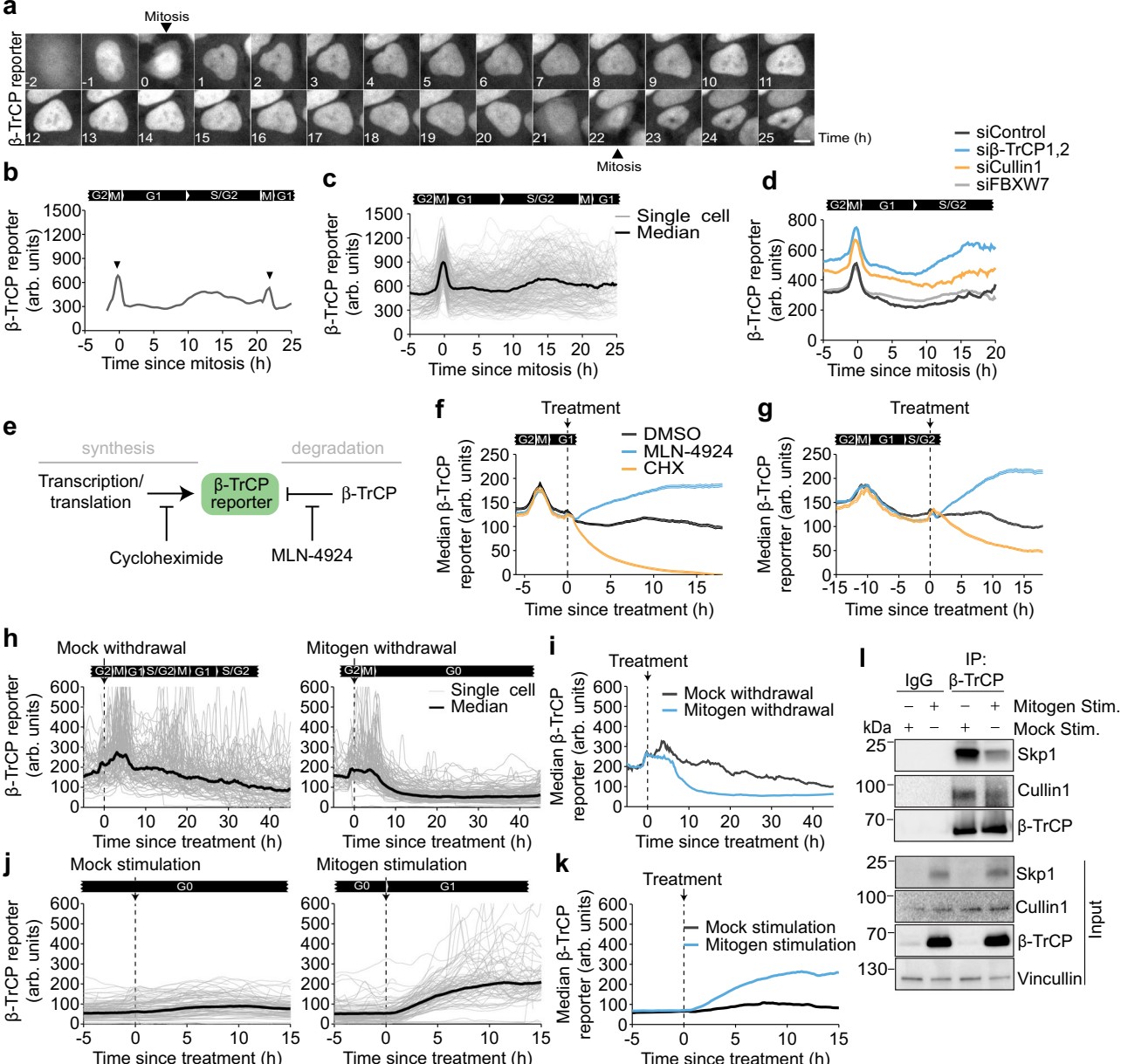

**Fig. 3 | β-TrCP activity is dynamically regulated during cell cycle entry and exit.**
**a** Image montage of a single-cell expressing the β-TrCP reporter. Images were taken at the indicated time (h) relative to mitosis. Scale bar is 10 μm. **b** Representative trace of β-TrCP reporter levels in a single HeLa cell. Note the artificial increase in fluorescence intensity during mitosis (black arrow) due to cell rounding and nuclear envelope breakdown. **c** Single-cell and median β-TrCP reporter levels in HeLa cells aligned to mitosis. $N = 263$ cells. **d** Median β-TrCP reporter levels in HeLa cells treated with either siControl, siβ-TrCP1,2, siCullin1, or siFBXW7. **e** Schematic diagram showing that β-TrCP reporter levels are controlled through a balance of synthesis and degradation. Treatment of either cycloheximide (CHX) or MLN-4924 can inhibit these processes. **f, g** Median β-TrCP reporter levels in HeLa cells treated with either DMSO, MLN-4924 (3 μM), or cycloheximide (100 μg/mL). Only cells that

were treated in either G1 (**f**) or S/G2 phase (**g**) of the cell cycle were analyzed. **h, i** Single-cell (**h**) and median (**i**) β-TrCP reporter levels in MCF-10A cells after mock or mitogen withdrawal. Only cells that were in S/G2 phase at the time of mitogen withdrawal were analyzed. $N = 93$ (Mock withdrawal) and 100 (Mitogen withdrawal) cells. **j, k** Single-cell (**j**) and median (**k**) β-TrCP reporter levels in quiescent MCF-10A cells that were either mock or mitogen stimulated. $N = 100$ cells per condition. **l** MCF-10A cells were starved for 48 h to induce quiescence, and then either mock or mitogen stimulated for 2 h before cell lysates were collected. Active complex formation is assessed by relative association of the SCF component Cullin1 and Skp1 with β-TrCP. Representative blot from $n = 2$ experiments. Source data for all figure panels are provided as a Source Data file.

for many cellular processes. To assess the sensitivity of the developed reporter towards β-TrCP1 and β-TrCP2, we knocked them down individually and in combination. We found that both the nuclear and the cytoplasmic β-TrCP reporter signal is primarily responsive to β-TrCP2 knockdown and moderately responsive to β-TrCP1, supporting previous observations that β-TrCP2 is the dominant paralog in most cells (Supplementary Fig. 3h).

Our β-TrCP reporter was based on a fragment of a non-canonical substrate. To determine if the dynamics of β-TrCP activity we observed were relevant specifically to non-canonical substrates, we designed another reporter utilizing the canonical β-TrCP substrate Emi1 (alias: FBXO5)[40]. We chose aa1–173 of Emi1 which contains the canonical β-TrCP degron but excludes the APC/C degron (Supplementary Fig. 3i). We hypothesized that since the β-TrCP-mediated degradation of Emi1

is dependent on phosphorylation[41], we could mutate the serine residues within the degron sequence to aspartic acid to induce constitutive β-TrCP-mediated degradation. We therefore mutated the "DSGYSS" of the wild-type Emi1 degron (Emi1-WT) to "DEGYSE" (Emi1-EE; Supplementary Fig. 3j). To assess the relative dynamics of the β-TrCP reporters derived from both the non-canonical (CDC25B; β-TrCP reporter) and canonical (Emi1-WT and Emi1-EE degrons) substrates, we made stable HeLa cells expressing both reporters fused to different fluorescent proteins (mVenus and mCherry respectively). Using live-cell imaging, we found near identical traces for the β-TrCP reporter and Emi1-EE degrons (Supplementary Fig. 3k). The Emi1-WT degron was only degraded to the same level as the non-canonical and Emi1-EE degrons during mitosis when it is known to be phosphorylated by PLK1 prior to being ubiquitinated by β-TrCP. Thus, our β-TrCP reporter reflects the dynamic regulation of both canonical and non-canonical substrates.

Given that the transcription of the β-TrCP reporter is driven by a constitutive promoter (Supplementary Fig. 1g), the levels of the β-TrCP reporter we observed in cells are the result of a competition between synthesis and β-TrCP-mediated degradation. To better understand how the β-TrCP reporter dynamics we observed in cells were shaped specifically by β-TrCP activity, we pre-imaged cells expressing the β-TrCP reporter to establish the history and cell cycle phase of each cell and then treated them with either MLN-4924, or cycloheximide (Fig. 3e). We observed a rapid increase in β-TrCP reporter levels following the addition of MLN-4924 regardless of whether a cell was in G1 or S/G2 phase. Furthermore, the rate at which the reporter accumulated reflects its synthesis rate (Fig. 3f, g and Supplementary Fig. 4a–c). Notably, following MLN-4924 treatment the β-TrCP reporter reached similar levels to the mutant β-TrCP reporter which lacks a functional degron sequence, indicating that β-TrCP is the primary regulator of the reporter degradation (Supplementary Fig. 4d). Opposite to our observations with MLN-4924, we observed a rapid decrease in reporter levels following the addition of cycloheximide, and the rate at which the reporter decreased reflects β-TrCP activity. Notably, we observed a more rapid degradation of the β-TrCP reporter following cycloheximide treatment when cells were in G1 phase (half-life of 2.75 h) compared to S/G2 phase (half-life of 9 h) (Supplementary Fig. 4e). These results confirm our earlier observation from unperturbed cells expressing the β-TrCP reporter and show that β-TrCP activity is higher in G1 phase compared to S/G2 phase.

We next investigated whether β-TrCP was regulated during exit to quiescence or re-entry into the cell cycle. We found that after mitogen withdrawal β-TrCP reporter levels fell rapidly after the last mitosis before cells entered G0, indicating that β-TrCP activity increases during G0 (Fig. 3h, i). This observation was not the result of reduced transcription or translation of the reporter itself as treatment of quiescent cells with MLN-4924 resulted in a rapid and immediate increase in β-TrCP reporter levels (Supplementary Fig. 4f). Interestingly, when we stimulated quiescent cells with mitogens (EGF, Insulin, and 5% horse serum), we observed a rapid increase in β-TrCP reporter levels, indicating mitogen signaling rapidly inhibits β-TrCP activity (Fig. 3j, k and Supplementary Fig. 4g). Thus, our data indicates that β-TrCP is relatively more active during quiescence and inhibited by mitogen signaling during cell cycle entry. Strikingly, and apparently contradictory to these conclusions, we found that β-TrCP protein levels were actually reduced during quiescence and elevated upon mitogen stimulation (Fig. 3l). To better understand how reduced β-TrCP protein levels could result in increased β-TrCP activity, we performed co-immunoprecipitation experiments between β-TrCP and the core SCF subunits SKP1 and Cullin1 after serum withdrawal. We found that despite lower total β-TrCP protein levels, more of the β-TrCP protein was in complex with the SCF compared to cells stimulated with mitogens for only three hours (Fig. 3l and Supplementary Fig. 4h). Taken together, our data indicates that mitogen signaling regulates

β-TrCP activity in part by inhibiting complex formation between β-TrCP and the core SCF complex.

## β-TrCP activity is regulated at the level of SCF complex formation

To further explore the observation that β-TrCP protein levels did not reflect changes in β-TrCP activity during quiescence and cell cycle re-entry, we sought to measure β-TrCP protein levels and activity in the same single cells. We first sought to derive β-TrCP activity in single cells directly by using changes in the fluorescent intensity of the β-TrCP reporter, a method described previously that relies on the reporter transcription being driven by an unregulated and constitutive promoter and the reporter protein degradation being driven primarily by the ubiquitin ligase[37]. To determine the rate of synthesis of the β-TrCP reporter, we treated cells with MLN-4924 to inhibit the β-TrCP-mediated ubiquitination and degradation of the reporter and measured the slope of the increase in reporter levels, which is primarily being regulated by the constitutive promoter driving reporter transcription (Supplementary Fig. 5a, b). Using the experimentally measured synthesis rate of the β-TrCP reporter, we converted β-TrCP reporter levels to β-TrCP activity for a single cell (Fig. 4a). As described above in our earlier analysis (see Fig. 3b, c), this single cell had relatively high levels of β-TrCP activity in G1 phase. However, once this cell entered S phase, which was determined by measuring APC/C activity in the same cell (Supplementary Fig. 5c), the β-TrCP activity dropped to a relatively lower level. When we expanded this analysis to 200 cells (Fig. 4b), we found a similar trend. These data together with our earlier analysis indicate that β-TrCP activity is high in G1 phase and then falls to a relatively lower level upon entry into S phase, but notably never fully inactivates.

Having established a method for measuring β-TrCP activity in single cells, we next sought to investigate the relationship between β-TrCP protein levels and β-TrCP activity. We performed time-lapse imaging to first measure β-TrCP activity in single cells, and after imaging, we fixed the cells and performed immunofluorescence using a β-TrCP antibody (Supplementary Fig. 5d, e). Using this approach, we plotted the β-TrCP activity as a function of β-TrCP protein levels in several hundred cells (Fig. 4c, d). We did not observe any correlation between β-TrCP protein levels and β-TrCP activity at the single-cell level. We found a similar lack of a correlation not only in MCF-10A cells, which are immortalized normal breast epithelial cells, but also in MCF7 and MDA-MB-231 cells, which are different grades of breast cancer cells.

In addition to looking at the regulation of β-TrCP activity in single cells, we also investigated how β-TrCP activity is regulated in different cell types. β-TrCP is a known oncogene involved in the establishment and progression of cancer. It has been found to be overexpressed in pancreatic, lung, ovarian, and breast cancer, and high expression of β-TrCP is linked to an overall poor prognosis[13,23,33,42,43]. To understand how β-TrCP activity may be altered in different cancers with elevated expression of β-TrCP, we stably expressed the β-TrCP reporter in cell lines with variable expression levels of β-TrCP, as measured by western blotting, ranging from relatively low levels (U2OS) to relatively high levels (HeLa, NCI-H460) (Supplementary Fig. 5f). We again found no correlation between the levels of β-TrCP protein and β-TrCP activity across cell lines (Supplementary Fig. 5g, h). For example, as noted in Supplementary Fig. 5f, MCF7 and MDA-MD-231 cells, which are both breast cancer cell lines, express similar levels of β-TrCP protein. However, MDA-MB-231 cells exhibited almost threefold higher levels of β-TrCP activity compared to MCF7 cells (Fig. 4d and Supplementary Fig. 5g, h). Considering that β-TrCP only functions as a ubiquitin ligase when it is part of a fully formed SCF complex, we sought to determine the relative amount of β-TrCP bound to other SCF components in these two cell lines. Similar to our experiments comparing quiescent cells with mitogen-stimulated cells, we found a greater amount of

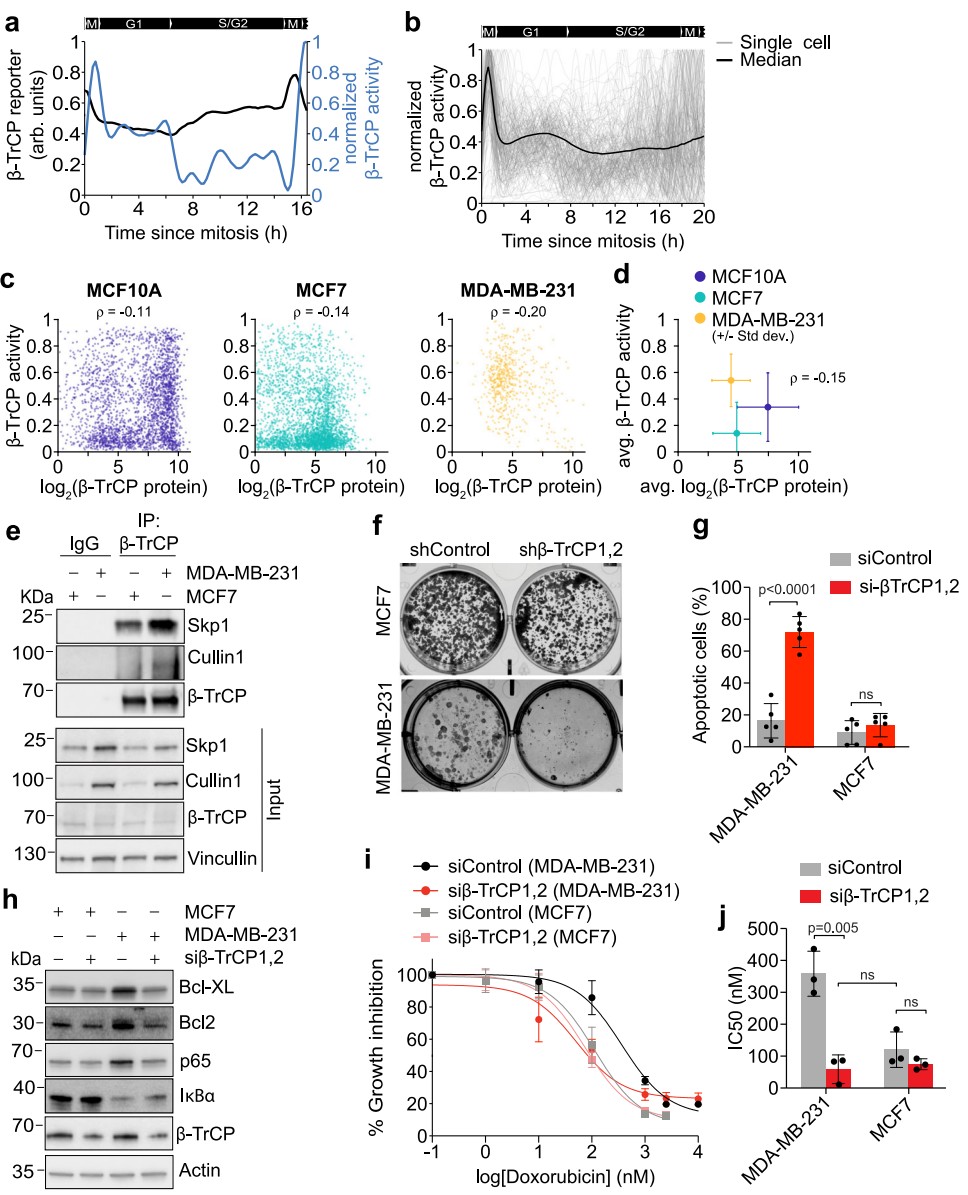

**Fig. 4 | β-TrCP activity is regulated at the level of SCF complex formation.**
**a** β-TrCP reporter levels (black) and β-TrCP activity (blue) for one MCF-10A cell.
**b** Single-cell and median traces of β-TrCP activity during the cell cycle. Traces were computationally aligned to time of mitosis. $N = 194$ cells. **c** Scatter plot showing the single-cell correlation of β-TrCP activity and β-TrCP protein levels. Following live-cell imaging, MCF-10A, MCF7, and MDA-MB-231 cells stably expressing the β-TrCP reporter were fixed and stained for β-TrCP. **d** Scatter plot showing the mean β-TrCP activity and β-TrCP protein levels from the single-cell data in (**c**) Error bars represent standard deviation. ρ; correlation coefficient. Representative plots from $n = 3$ experiments. **e** Relative association of β-TrCP with the SCF components Cullin1 and SKP1 in MCF7 and MDA-MB-231 cells. Whole-cell lysates were immunoprecipitated with anti-β-TrCP. Resulting immunoprecipitates were resolved and stained with indicated antibodies. Representative blot from $n = 3$ experiments. **f** Colony formation assay in MCF7 and MDA-MB-231 cells treated with either control shRNA or

β-TrCP1,2 shRNA. Representative image from $n = 3$ experiments. **g** Bar graph showing the mean percentage of Annexin-V5 positive cells from Supplementary Fig. 5j. Error bars represent SD from $n = 5$ experiments. *P* values were calculated using a two-way ANOVA and Sidak's multiple comparisons test. n.s. not significant ($p = 0.94$). **h** Immunoblotting of NF-kB target genes (Bcl-XL, Bcl2, p65) in MCF7, MDA-MB-231, and MDA-MB-231 cells treated with β-TrCP1,2 siRNA. Representative blot from $n = 3$ experiments. **i** Percent growth inhibition of MCF7 and MDA-MB-231 cells, treated with different doses of doxorubicin. Cells were treated with either control siRNA or β-TrCP1,2 siRNA. Error bars represent SEM from $n = 3$ experiments. **j** Bar graph showing the mean IC50 of the dose–response curves in (**i**). Error bars represent SD from $n = 3$ experiments. *P* values were calculated using a one-way ANOVA. n.s. not significant ($p = 0.69$ and $p = 0.88$ respectively). Source data for all figure panels are provided as a Source Data file.

β-TrCP co-immunoprecipitated with Skp1 and Cullin1 in MDA-MB-231 cells compared to MCF7 cells (Fig. 4e). This data indicates that despite near equal amounts of β-TrCP protein, a greater proportion of that protein forms an active SCF-β-TrCP complex in MDA-MB-231 cells compared to MCF7 cells.

In order to determine whether the difference in β-TrCP activity we observed between MCF7 and MDA-MB-231 cells was physiologically relevant, we sought to measure the relative reliance on β-TrCP activity

between these two cell lines. To answer this question, we knocked down β-TrCP1,2 in MCF7 and MDA-MB-231 cells (Supplementary Fig. 5i). Interestingly, we found MDA-MB-231 failed to grow upon knockdown of β-TrCP1,2, whereas MCF7 cells grew similar to control upon knockdown of β-TrCP1,2 (Fig. 4f and Supplementary Fig. 5j). By staining for Annexin V and assessing the levels of PARP1 and Caspase 3 cleavage, we found MDA-MB-231 cells undergo apoptosis upon knockdown of β-TrCP1,2, whereas MCF7 cells do not undergo

apoptosis (Fig. 4g and Supplementary Fig. 5k, l, m). Thus, MDA-MB-231 cells, which have relatively high levels of β-TrCP activity, are sensitive to β-TrCP1,2 knockdown and undergo apoptosis, demonstrating that these cells rely more heavily on β-TrCP activity for survival than cell lines that have relatively low levels of β-TrCP activity. This reliance on β-TrCP activity for cell survival is explained by increased basal levels of NF-kB signaling in MDA-MB-231 compared to MCF7 cells, as shown by increased levels of the NF-kB target genes Bcl-XL and Bcl2[44,45] (Fig. 4h and Supplementary Fig. 5n). These increases in protein levels could be rescued by knocking down β-TrCP in MDA-MB-231 cells but not MCF7 cells. Finally, we found that knocking down β-TrCP1,2 in MDA-MB-231 cells significantly sensitized them to doxorubicin treatment (Fig. 4i) with a nearly fivefold decrease in the IC50 (Fig. 4j), while knocking down β-TrCP1,2 in MCF7 had limited effect. Thus, MDA-MB-231 cells rely more heavily than MCF7 cells on elevated levels of β-TrCP activity to increase NF-kB signaling, which in turn increases the levels of pro-survival and anti-apoptotic genes and makes them less sensitive to chemotherapy.

## High-throughput screening approach using the β-TrCP reporter

In addition to revealing the dynamic regulation of β-TrCP activity, we reasoned that our β-TrCP reporter could be used as a screening tool to identify new regulators of β-TrCP activity. To facilitate a high-throughput screening (HTS) approach, we developed a plate reader-compatible version of our β-TrCP reporter by replacing the mVenus tag with a NanoLuc luciferase (NLuc) tag[46] (Fig. 5a). We chose to use MDA-MB-231 cells because they showed high levels of β-TrCP activity and would therefore exhibit the highest dynamic range in our initial screen. We scaled our assay to make it compatible with 1536-well plates and demonstrated in MDA-MB-231 cells, using MLN-4924 and cyclo-heximide as positive controls for both inhibitors and activators respectively, that the assay was sufficiently reproducible for HTS. As expected, we found increased luminescence in cells treated with MLN-4924 and decreased luminescence in cells treated with cycloheximide (Fig. 5b), consistent with our earlier validation experiments with the β-TrCP-mVenus reporter (see Fig. 1f, g). We screened the NCATS Pharmaceutical Collection (NPC 2.0) and the Mechanism Interrogation PlatE (MIPE 5.0) libraries[47], which together contain ~5000 compounds with known mechanisms of action, to identify small-molecule modulators of β-TrCP activity (Fig. 5c, d). We identified several small molecules that were categorized as potential inhibitors or activators of β-TrCP activity (Supplementary Fig. 6a). As expected, among the initial hits were several different proteasome inhibitors, a ubiquitin-activating enzyme (UAE) inhibitor, and a protein translation inhibitor, demonstrating a high-success rate for on-target hits. Unexpectedly, we identified several kinase inhibitors involved in receptor-tyrosine kinase (RTK) signaling, including several EGFR inhibitors, as potential β-TrCP activators (Fig. 5e and Supplementary Fig. 6a). The high penetrance of RTK signaling components in our screen strongly supports a hypothesis that RTK signaling is a key regulatory pathway controlling β-TrCP activity and is consistent with our earlier observation that β-TrCP activity was inhibited by mitogen stimulation (see Fig. 3h–l).

To further explore this hypothesis, we chose the multi-RTK inhibitor sunitinib[48] for further validation as it is an approved drug to treat multiple cancers including Renal and GI[49,50]. We confirmed that sunitinib selectively enhances β-TrCP activity, as sunitinib had no effect on a reporter for the similar SCF^Skp2 ubiquitin ligase nor the mutant β-TrCP reporter (Fig. 5f, g and Supplementary Fig. 6b). Consistent with sunitinib enhancing β-TrCP activity, treatment with sunitinib resulted in a rapid decrease in β-TrCP reporter levels within 1 h as assessed by live-cell imaging (Fig. 5g), as well as the levels of the endogenous β-TrCP substrates DYRK1A, CDC25B, IκBα, PFKFB3, and Emi1 as assessed by western blotting (Fig. 5h and Supplementary Fig. 6c). We did not observe any change in β-TrCP protein levels following sunitinib

treatment, indicating that sunitinib must modulate β-TrCP activity directly. We again immunoprecipitated β-TrCP and found increased interaction of β-TrCP with Skp1 upon treatment with sunitinib, suggesting the increased β-TrCP activity we observed is due to an increase in the assembly of β-TrCP into a functional SCF complex (Fig. 5i). Finally, we tested the effect of sunitinib on the chemosensitivity of MDA-MB-231 cells to doxorubicin. We found that treatment with sunitinib rendered MDA-MB-231 cells less sensitive to doxorubicin (Fig. 5j; compare black line to red line) and this protective effect requires β-TrCP (compare gray line to green line). Thus, our data demonstrate that inhibiting RTK signaling in MDA-MB-231 cells enhances β-TrCP activity by increasing the association of β-TrCP with the SCF complex, rendering them less sensitive to chemotherapy.

## Discussion

The ubiquitin ligase β-TrCP is a critical regulator of many biological processes including the cell cycle and inflammatory signaling[13]. In order to gain a more detailed understanding of the dynamic regulation of β-TrCP in live cells, we developed a genetically encoded fluorescence reporter for β-TrCP activity. To study β-TrCP activity directly without the influence of the various kinases that are required to phosphorylate canonical β-TrCP substates, we designed our reporter based on a non-canonical substrate, CDC25B, which does not require prior phosphorylation in order to bind to β-TrCP[18]. Our validation and characterization of this reporter revealed insights into how β-TrCP activity is regulated in live cells throughout the cell cycle, and the high-throughput assay we developed provides a platform for future drug discovery. Unlike other ubiquitin ligases such as APC/C[37,51], SCF^SKP2 [30], or CRL4^Cdt2 [32], which are all capable of completely degrading their substrates, we found that β-TrCP primarily functions to maintain its substrates at discreet steady-state levels. This was true not only for our engineered reporter, but also for canonical substrates such as Emi1, which were never fully degraded, but rather changed from high levels to low levels following mitosis (see Supplementary Fig. 3k). This observation suggests that elevated activity of β-TrCP in different cancer cells may change the steady-state levels of its substrates, thereby altering the balance of fate-determining signaling pathways.

Our cell cycle analysis of β-TrCP activity revealed that β-TrCP is highly active during quiescence or G0 phase, moderately active in G1 phase, and least active in S/G2 phase (Fig. 5k). This observation somewhat contradicts the current model which says β-TrCP should be mostly active in S/G2 phase because most of the well-known β-TrCP substrates are degraded in S phase (e.g. CDC25A), G2 phase (e.g. Wee1), or mitosis (e.g. Emi1)[14]. This raises the questions of what role does β-TrCP play, and which substrates are degraded, during quiescence. Given that substrate ubiquitination is controlled both by the level of β-TrCP activity as well as the activity of various kinases, our β-TrCP reporter does not necessarily reflect the protein levels of canonical substrates at a given point in the cell cycle. Rather, the reporter reflects the potential for these canonical substrates to be degraded upon kinase activation. One potentially critical β-TrCP substrate during quiescence is the mTOR inhibitor DEPTOR[52]. Previous work has shown that DEPTOR is degraded upon nutrient stimulation via AKT-mediated phosphorylation and subsequent ubiquitination by β-TrCP, thus permitting the activation of mTOR. Our data indicates that β-TrCP activity is also reduced after mitogen stimulation, thereby limiting the amount of DEPTOR degradation, and by extension mTOR activation, during cell cycle entry. Why cells may want to limit DEPTOR degradation during cell cycle entry is not known, but we speculate that cells may use high β-TrCP activity during quiescence to maintain the relative levels of DEPTOR and mTOR in order to maintain proper sensitivity to mitogens or nutrients when they are made available.

As mentioned above, high β-TrCP protein levels in a tumor are associated with a poor clinical prognosis, and β-TrCP protein levels

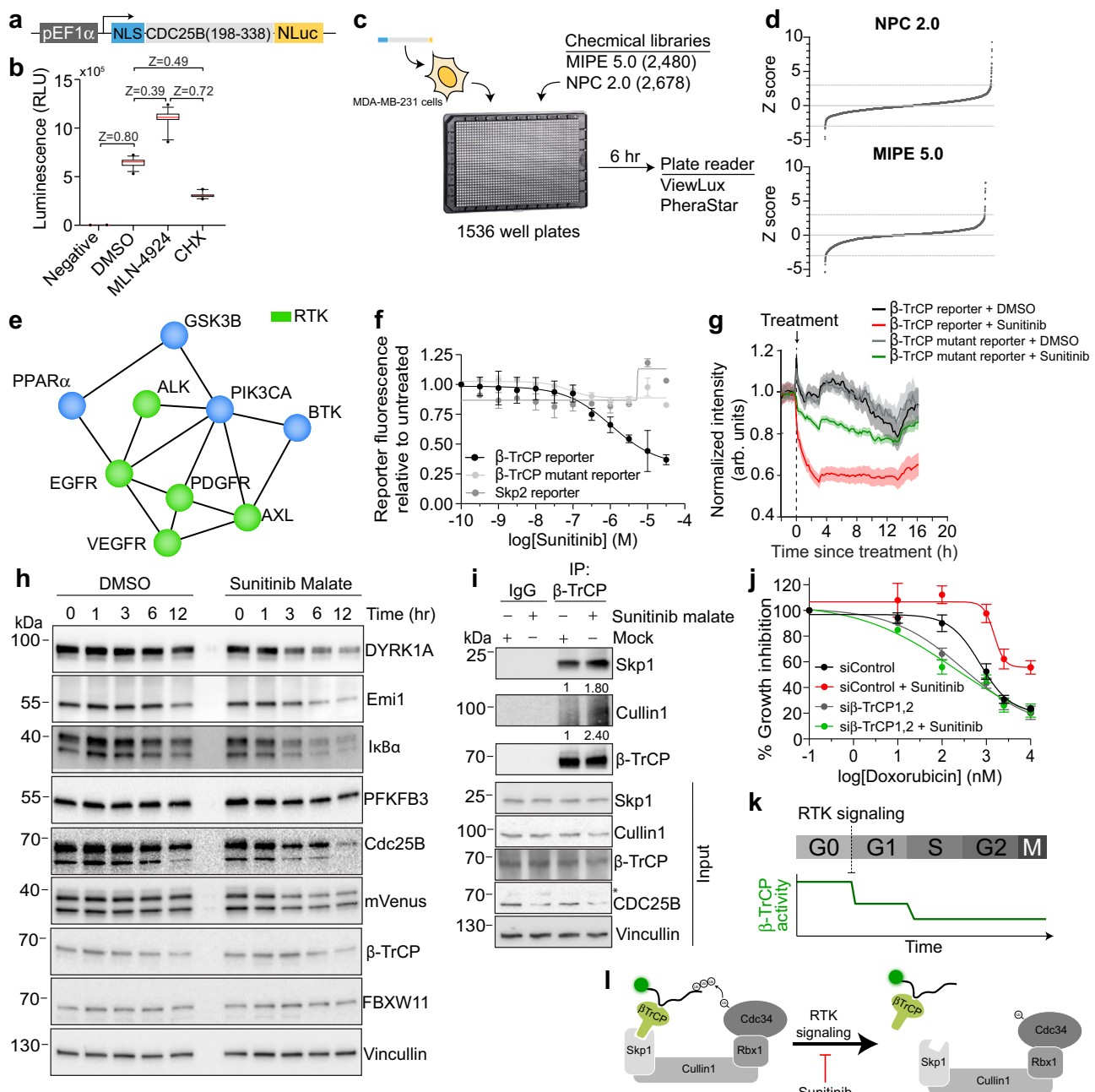

**Fig. 5 | High-throughput screening approach using the β-TrCP reporter.**
**a** Schematic showing the β-TrCP reporter fused to NanoLuc luciferase (NLuc).
**b** Relative luciferase activity of MDA-MB-231 cells stably expressing β-TrCP reporter treated with DMSO, MLN-4924, or cycloheximide (CHX) for 6 h. Graph is a box-and-whisker plot where the red line represents the median, the box represents the inter-quartile range, and the whiskers represent 5–95 percentiles. Dots represent indivi-dual data points beyond these percentiles. $n = 32$ replicates. **c** Workflow of the high-throughput screening (HTS) strategy. MDA-MB-231 cells stably expressing the β-TrCP-NLuc reporter were seeded in 1536-well plates. The initial screen used the MIPE 5.0, and NPC 2.0 libraries comprising over 5000 approved and investigational drugs. After 6 h of treatment, luminescence was measured using a PHERAStar FSX micro-plate reader (BMG Labtech). **d** Scatter plot summarizing luminescence of MDA-MB-231 cells expressing β-TrCP-NLuc reporter treated with compounds from MIPE 5.0 and NPC 2.0 library as described in (**c**) Each dot represents the percent activity of each small-molecule compound tested normalized to neutral and negative controls. Shaded regions highlight compounds demonstrating greater than 50% change relative to DMSO-treated controls. **e** STRING pathway analysis of proteins whose inhibitors were found to activate β-TrCP activity in our screen. Nodes were colored green if they are a receptor-tyrosine kinase (RTK). **f** Relative reporter levels of

MDA-MB-231 cells expressing either β-TrCP-mVenus reporter, mutant β-TrCP reporter, or Skp2-mCherry reporter treated with increasing doses of Sunitinib. Error bars represent SEM from $n = 2$ repeats. **g** Single-cell traces of β-TrCP-mVenus reporter levels in MDA-MB-231 cells treated with Sunitinib at the indicated time. Data is median ± 95% confidence intervals. **h** Exponentially growing MDA-MB-231 cells were treated with Sunitinib (3 µM) for the indicated times. Whole-cell lysates were resolved in SDS-PAGE and immunoblotted for the indicated proteins. Representative blot of $n = 3$ independent experiments. **i** Immunoprecipitation of β-TrCP with the SCF complex components Skp1 and Cullin1 in MDA-MB-231 cells treated with either DMSO or Sunitinib. Quantification of enrichment of bands compared to control is shown beneath the relevant bands. Representative blot of $n = 3$ independent experiments. **j** CCK-8 assay to determine percent growth inhibition of MDA-MB-231 cells treated with either control siRNA or β-TrCP1,2 siRNA. Cells were treated with different doses of doxorubicin with or without 5 µM sunitinib. Error bars represent SEM from $n = 4$ experiments. **k** Model showing relative β-TrCP activity across quiescence (G0) and different cycling phases. **l** Discovery of RTK signaling as a modulator of active SCF- β-TrCP complex. Sunitinib malate promotes the association of β-TrCP with core SCF complex components. Source data for all figure panels are provided as a Source Data file.

have been proposed as a prognostic marker[53]. However, our studies using the β-TrCP reporter demonstrate that the absolute β-TrCP protein levels may not be clinically useful as they do not necessarily indicate that a cell has elevated β-TrCP activity. This was the case for genetically similar cells within a population as well for different cancer cell lines. Most strikingly, we made the same observation within the same cell, where β-TrCP was highly active in quiescence, and then was rapidly inhibited upon stimulation with growth factors. Mechanistically, this is because β-TrCP must bind to the rest of the SCF complex to be a fully functional E3 ubiquitin ligase, indicating that β-TrCP activity is largely regulated through SCF complex formation rather than by expression levels. Our data demonstrate that this complex formation is likely inhibited, at least in part, by receptor-tyrosine kinase signaling, as treatment with the multi-RTK inhibitor sunitinib increased binding between β-TrCP and the SCF component Skp1 (Fig. 5l). This hypothesis is also supported by our observation that mitogen treatment, which included the RTK ligands EGF and Insulin resulted in reduced binding between β-TrCP and both Skp1 and Cullin1. Further work is now needed to understand how RTK signaling controls SCF-β-TrCP complex formation. The mechanism could involve direct post-translational modifications of either β-TrCP itself or specific SCF components or it could involve regulation of proteins like CAND1[29], which reportedly controls the shuttling of F-box proteins in and out of the SCF complex.

In addition to the fluorescent β-TrCP reporter, which is useful for single-cell and quantitative analysis, we developed a luminescent β-TrCP reporter, which is useful for high-throughput screening. We screened two chemical libraries comprised of drugs with known molecular targets in order to identify proteins or signaling pathways that regulate β-TrCP activity. In the future, such an approach can be used to identify a specific small-molecule inhibitors of β-TrCP activity, which could be useful for the treatment of specific cancers[54,55]. The assay we designed, as well as the follow-up counter assay employing reporters for SCF$^{Skp2}$ and the mutant β-TrCP reporter, will also be useful for discovering molecules such as β-TrCP activators, pan-SCF modulators, or proteasome inhibitors, among others. These compounds could prove useful both for research as well as for clinical use. Our study will serve as a roadmap for future work uncovering the precise regulatory mechanisms controlling β-TrCP activity in both normal and cancer cells.

## Methods

### Cell culture
Human MCF-10A (CRL-10317), MCF7 (HTB-22), MDA-MB-231 (CRM-HTB-26), MDA-MB-468 (HTB-132), U2OS (HTB-96), HeLa (CRM-CCL-2), and NCI-H460 (HTB-177) cancer cell lines and transformed human embryonic kidney (HEK-293T; CRL3216) cells were obtained from ATCC. MCF7, MDA-MB-231, HeLa, MDA-MB-468, and HEK-293T cells were cultured at 37 °C in Dulbecco's modified Eagles medium (DMEM) (Gibco, Life Technologies, Carlsbad CA, USA) containing 10% fetal bovine serum (FBS; Gibco). MCF-10A cells were cultured in the following full-growth media: phenol red-free DMEM/F12 (Invitrogen) supplemented with 5% horse serum, 20 ng/mL EGF, 10 μg/mL insulin, 500 μg/mL hydrocortisone, 100 ng/mL cholera toxin, and 1% P/S. All tissue culture media were supplemented with 2 mM L-glutamine, 25 μg/ml streptomycin and 25 U penicillin (Gibco). Cells were cultured in a humidified atmosphere with 5% $CO_2$ at 37 °C.

### Constructs and stable cell lines
CSII-pEF1a-H2B-mTurquoise, CSII-pEF1a-mCherry-Geminin(aa1–110), and CSII-pEF1a-Cdt1-mVenus (Skp2 reporter) were described previously[30,37,38]. CDC25B (aa198-338), was synthesized as a G-block (IDT) and cloned along with m-Venus and CSII-pEF1a (digested with Bam H1 and EcoR1) using the Gibson assembly method[56]. A similar strategy was used to make the m-Venus tagged variants of CDC25B as

well as in preparation of Emi1-WT and the phospho-mimetic mutant (Emi1-EE). FLAG- β-TrCP1 and ΔF- FLAG- β-TrCP1 is a kind gift from Dr. Yongchao Zhao (Zhejiang University School of Medicine, Hangzhou, China). His-Ubiquitin was a kind gift from Dr. Manas Kumar Santra (National center for cell science, Pune, India). Lentivirus-transduced cells were sorted on a BD Biosciences FACS-Aria Fusion to obtain pure populations expressing the desired fluorescent reporters. For the luminescent reporter system, cells were infected with high-titer virus containing Nluc tagged β-TrCP reporter, infection media changed after 24 h and then selected with puromycin until all the cells in the control (Uninfected cells) condition died.

The Skp2 reporter used in this study is part of the original FUCCI reporter system and is comprised of the plasmid CSII-pEF1a-Cdt1(aa30–120)-mVenus. This fragment of Cdt1 is not functional but contains a Skp2 degron. While the fragment lacks a phosphorylated threonine at position 29, previous validation by Grant et al. has demonstrated that the linker connecting Cdt1(aa30–120) to the fluorescent protein contains a phospho-mimetic glutamate residue that ensures the degron is constitutively degraded when SCF$^{Skp2}$ is active[32].

### Lentivirus construction and production
cDNA encoding mVenus-hCDC25B (aa198-338), mVenus-Emi1-WT (aa1-173), or mVenus-Emi1-EE (aa1-173) were cloned into a CSII-EF-MCS vector. The plasmid was co-transfected with lentivirus packaging plasmids (pCAG-HIVgp) and the VSV-G- and Rev-expressing plasmid (pCMV-VSV-G-RSV-Rev) into HEK 293T cells. High-titer viral solutions for mVenus-CDC25B (aa198-338), mVenus-Emi1-WT (aa1-173), mVenus-Emi1-EE (aa1-173), mTurquoise -H2B, mCherry-Geminin (aa1−110) were prepared and used for co-transduction into the indicated cell lines.

### Inhibitors
Cells were treated with either vehicle (DMSO) or 3 μM MLN-4924 (Calbiochem, USA), 100 μg/ml cycloheximide (Sigma, USA), 5 μM MG132 (Calbiochem, USA), or 5 μM Sunitinib malate for the indicated time points. The cells were either harvested after treatment and whole-cell extracts were prepared or imaging was continued.

### siRNA transfection/shRNA
Indicated cells were transfected using Dharmafect 1 (Horizon Discovery, Ltd) according to the manufacturer's instructions. The following siRNAs were used: On-Target plus control siRNA (nontargeting, Dharmacon), On-Target plus pooled set of four siRNAs for β-TrCP1 (L-003463-00-0005) and β-TrCP2 (L-003490-00-0005), On-Target plus pooled set of four siRNA for FBXW7 (L-004264-00-0005), and Cullin1 (L-004086-00-0005) at final concentrations of 20 nM unless noted. Six hours post transfection, cells were washed with full-growth medium and then imaging was started 18 h later. For shRNA mediated knockdown, cells were infected with lentivirus containing either control or β-TrCP1,2 shRNA, and media was changed after 24 h followed by selection with antibiotic.

### Cell proliferation assay
The proliferation rate of different cells was determined using a cell counting kit-8 (CCK-8, Dojindo, Shanghai, China) according to the manufacturer's instructions. In brief, indicated cells were seeded on a 96-well plate at the density of 5000 cells per well. 24 h after seeding, cells were transfected with either control or β-TrCP1,2 siRNA. Cells were treated with increasing concentrations of different drugs or vehicle (DMSO) for 48 h. CCK-8 was added, and the absorbance at 450 nm was measured using a microplate reader (Tecan plate reader).

### Colony formation assay
Five thousand cells (transduced) were seeded in 35-mm culture dish and allowed to grow for 12–15 days to form the colonies. Cells were

then fixed with 4% formaldehyde, washed with PBS, and finally stained with 0.05% crystal violet. Plates were photographed, and the representative images were shown. Colonies were counted using Alpha View software and plotted as percent compared to shControl (set as 100%).

### Cell apoptosis analysis with PI/Annexin V double staining

MCF7 and MDA-MB-231 cells were harvested 48 h after transfection with either control or β-TrCP1,2 2 nM siRNA. Harvested cells washed in PBS, after which $1 \times 10^6$ cells were resuspended and stained using Dead Cell Apoptosis Kit with Annexin V FITC and PI (V13242, Thermo Fisher) according to manufacturer's instructions and analyzed using flow cytometry (BD FACSCanto II; BD Biosicences). The intensity of FITC/Annexin V fluorescence was analyzed using FlowJo V10 software (FlowJo LLC) and presented on the x-axis, while PI was plotted on the y-axis. FITC/PI represented living cells, FITC+/PI indicated early apoptotic cells, FITC+/PI+ reflected late apoptotic cells and FITC/PI+ depicted necrotic cells.

### High-throughput SCF-β-TrCP-NLuc reporter assay

MDA-MB-231 cells stably expressing the β-TrCP-NLuc reporter were maintained in growth media (DMEM; ThermoFisher Scientific) supplemented with 10% FBS (ThermoFisher Scientific), 1× penicillin–streptomycin (ThermoFisher Scientific) and 1 μg/mL puromycin (Sigma). Prior to screening, cells were trypsinized, counted and seeded into 1536-well, solid bottom, tissue culture-treated plates (Greiner) at a density of $2.75 \times 105$ cells/mL in 5 μL of growth media lacking puromycin. Following overnight incubation, 23 nL of compound and controls (neutral control, DMSO or positive controls, MLN-4924 or Cycloheximide (CHX) at final concentrations of 3 μM with final DMSO concentrations of 0.46%) were delivered using Kalypsys pintool transfer. The assay plates were then incubated for 6 h at 37 °C with 5% $CO_2$ before 2 μL of 3.5X NanoGlo Luciferase Detection Reagent (Promega) was added, as described by the manufacturer's instructions, using a BioRaptr 2 FRD Dispenser (Let's Go Robotics) and luminescence intensity was measured using a PHERAstar FSX plate reader (BMG Labtech) with luminescent module after 10 min at room temperature. For primary high-throughput screening (HTS), each compound was tested at a single concentration (13 μM final in DMSO) and screening data were normalized and percent activity was computed using the median values of the neutral control (32 wells containing cells and DMSO) and the negative control (32 wells containing media alone and DMSO). Compounds demonstrating >50% inhibition or activation relative to mean DMSO controls were selected as inhibiting or activating hits, respectively. Hits were then re-plated in 11 pt. titration at 1:3 dilutions, re-tested and IC50 values were calculated based on curve fitting to normalized data, as described above. Concentration-response curves, Curve Class, and Efficacy were derived using in-house algorithms[57].

### Analysis of cell cycle profile by flow cytometry

Exponentially growing Parental cells and cells stably expressing the β-TrCP reporter were harvested and fixed in 95% ethanol. On the day of cell acquisition for flow cytometry, cells were washed with ice-cold PBS followed by staining with propidium iodide and immediately acquired in BD FACS Calibur.

### Quantitative real-time RT-PCR

RNA was isolated from cultured cells using RLT RNeasy 96 Qiacube HT Kit (Qiagen, 74171) and on-column DNA digest (Qiagen, 79254). Total RNA was extracted from cells and 1 μg total RNA was used to prepare cDNA. The expression of GAPDH was used to normalize the data (Fwd: 5′ TCAGCAAAGCCCTGAGTCAG 3′, Rev: 5′ CATCTCTGTCGTCGTCCT CG 3′). Bcl-XL primers (Fwd: 5′ AGCTTGGATGGCCACTTACC 3′ and Rev: 5′ AAGAGTGAGCCCAGCAGAAC 3′) and Bcl2 primers (Fwd: 5′ GGGGTCATGTGTGTGGAGAG 3′ and Rev: 5′ CATCCCAGCCTCCGT

TATCC 3′) were used in this study. The experiment was repeated two times, and the fold change was calculated using the ΔΔCT method.

### Immunofluorescence

Cells were fixed in 4% paraformaldehyde, washed three times in PBS, permeabilized with 0.2% triton, and stained overnight at 4 °C with anti-β-TrCP antibody (sc366369). Primary antibodies were visualized using a secondary antibody conjugated to Alexa Fluor-647 and imaged with a Far-Red filter.

### Cell lysis and immunoblotting

Cell lysis was carried out largely as previously described[58]. Cells were harvested and washed twice with ice-cold phosphate buffer saline (PBS). Cells were then lysed with whole-cell lysis buffer (50 mM Tris pH7.4, 200 mM NaCl, 50 mM NaF, 1 mM Na3VO4, 0.5% Triton X-100 and protease inhibitor cocktail) in ice for 30 min. Lysates were centrifuged at high speed ($16,000 \times g$) and clear supernatants were transferred to new tubes. Protein concentration was measured by the BCA method as per manufacturer instructions (Thermo Scientific, USA, 23225). Samples were prepared in SDS sample buffer and run in SDS-PAGE with Tris-Glycine (Bio-Rad, USA) running buffer. Separated proteins were transferred onto PVDF membrane with turbo-transfer buffer (Bio-Rad, USA). Membranes were incubated with primary antibody overnight at 4 °C and were subsequently incubated with HRP conjugated secondary antibody (Anti-rabbit IgG, HRP-linked antibody, CST-7074, 1:10,000, anti-mouse IgG, HRP-linked antibody, CST-7076, 1:10,000, or mouse anti-goat IgG-HRP Secondary Antibody, SCBT, sc2354, 1:15,000) for 1 h at room temperature. Blots were developed by the chemiluminescence method (SuperSignal™ West Pico and Femto PLUS reagents, Thermo scientific).

The following antibodies were used in this study: anti-mVenus (MyBioSource, MBS448126, 1:1000), anti-β-TrCP (Abcam, ab71753, 1:800), anti-β-TrCP (SCBT, sc390629; used for IPs, 2 μg), anti-β-TrCP (CST, 4394, used for IF, 1:250), anti-β-Catenin (BD-610153, 1:2500), anti-Vinculin (V9131, Sigma, 1:5000), anti-FBXW7 (Abcam, ab109617, 1:800), anti-FLAG (Sigma, F3165, 1:2000), anti-Ubiquitin (SCBT, sc8017, 1:800), anti-Cullin1 (SCBT, sc17775, 1:800), anti-SKP1 (SCBT, sc5281, 1:700), anti-CDC25B (CST, 9525, 1:1000), anti-Emi1 (SCBT, sc365212, 1:800), anti-HSP90 (CST, 4877, 1:1000), anti-Bcl-XL (CST, 2764, 1:1000), anti-Bcl2 (CST, 3498, 1:1000), anti-p65 (CST, 8242, 1:1000), anti-IKBα (CST, 4812, 1:1000), anti-caspase 3 (CST, 14220, 1:1000), anti-Cullin1 (CST, 4995, 1:1000), anti-cleaved PARP (CST, 5625, 1:1000), anti-NEDD8 (CST, 2745, 1:1000), anti-FBXO31 (Bethyl, A302-047A, 1:800), anti-mouse IgG (SCBT, sc2025, used for IPs, 2 μg), anti-DYRK1A (SCBT, sc100376, 1:800), anti-PFKFB3 (MyBioSource, MBS9604769, 1:1000), anti-FBXW5 (MyBioSource, MBS9611762,1:1000), anti-FBXO25 (MyBioSource, MBS3017705, 1:1000), anti-FBXW11 (Thermo scientific, PA5-109715, 1:1000), anti-cMyc (Abcam, ab32072, 1:800), and anti-β-Actin (Abcam, ab6276, 1:5000).

### Immunoprecipitation

Immunoprecipitation was carried out as previously described[58]. Cells were lysed in cell lysis buffer (50 mM Tris pH7.4, 200 mM NaCl, 50 mM NaF, 1 mM Na₃VO₄, 0.5% Triton X-100, and protease inhibitor cocktail). Whole-cell lysate (400–800 μg of proteins) was used for co-immunoprecipitation using 2 μg of antibody in modified IP lysis buffer (50 mM Tris pH7.4, 200 mM NaCl, 50 mM NaF, 1 mM Na₃VO₄, 0.1% Triton X-100, and protease inhibitor cocktail). The mixture of protein and antibody was kept at 4 °C in a rotor with gentle rocking for 12–16 h. The following day this antibody and cell lysate cocktail was allowed to bind to protein G-agarose beads for 2 h at 4 °C with gentle rocking. The immunoprecipitates were eluted from the beads using Laemmli buffer for 3–5 min and boiled prior to resolving on SDS-PAGE. Three percent of the proteins taken in the immunoprecipitation experiment was used as input in all immunoprecipitation assays.

## Ubiquitination assay

HEK-293T cells were co-transfected with different combinations of β-TrCP-mVenus reporter, FLAG-β-TrCP, FLAG-ΔF-β-TrCP, and 6X-His-Ubiquitin to assess the in vivo ubiquitination of the β-TrCP reporter. Cells were harvested after 48 h of transfection with the addition of 5 μM MG132 4 h prior to harvesting the cells. Cells were harvested and lysed and 600 μg of whole-cell lysate was used for immunoprecipitation. The ubiquitylated proteins were purified under denaturing conditions using Ni-NTA beads and immunoblotted using an anti-mVenus antibody to assess relative ubiquitylated levels of the β-TrCP reporter in the presence and absence of β-TrCP or dominant-negative β-TrCP. For endogenous ubiquitin pull downs (see Supplementary Fig. 2c), cells were transfected as indicated and harvested after 48 h of transfection with the addition of 5 μM MG132 4 h prior harvesting. Six hundred micrograms of whole-cell lysate was immunoprecipitated using anti-ubiquitin antibody and resultant immunoprecipitate was resolved in SDS-PAGE and probed for indicated protein. For mVenus pull downs (see Supplementary Fig. 2d–f), cells were harvested after 48 h of transfection with the addition of 5 μM MG132 4 h prior to harvesting the cells. Cells were harvested and lysed and 800 μg of whole-cell lysate was immunoprecipitated using anti-mVenus antibody. The resultant immunoprecipitates were resolved in SDS-PAGE and probed for anti-ubiquitin antibody.

## Time-lapse microscopy

Ninety-six-well plates (Ibidi, #89626) were coated with collagen (30 μg/ml) for 3 h and then cells were plated 24 h prior to imaging in full-growth media. Time-lapse imaging was performed in 200 μL full-growth media. Images were taken in CFP, YFP, and RFP channels every 12 min on a Nikon Ti2-E inverted microscope with a 10 × 0.45 NA or 20 × 0.75 NA objective and using NIS Elements software. Total light exposure time was kept under 150 ms for each time point. Cells were imaged in a humidified, 37 °C chamber in 5% $CO_2$. Cell tracking and data analysis was done using custom MATLAB scripts[37].

## Image analysis

All image analyses were performed with custom MATLAB scripts as previously described[37,59]. Briefly, optical illumination bias was empirically derived by sampling background areas across all wells in an imaging session and subsequently used to flatten all images. This enabled measurement and subtraction of a global background for each image. Cells were segmented for their nuclei based on either Hoechst staining (fixed-cell imaging) or H2B-mTurquoise (live-cell imaging). Nuclear β-TrCP-mVenus and mCherry-Geminin signals were calculated as median nuclear intensity. β-TrCP activity calculations were derived as previously described for the APC/C[37]. Briefly, the changes in the levels of the β-TrCP reporter in a cell at a given time are the result of the synthesis rate and the degradation rate of the reporter. We directly measured the synthesis rate of the reporter in individual cells following inhibition of β-TrCP using the neddylation inhibitor MLN-4924 (Supplementary Fig. 5a), and we were able to measure the levels of the β-TrCP reporter in single cells over time. By taking the first derivative of the β-TrCP reporter and subtracting the synthesis rate, we were able to calculate the degradation rate of the reporter, which is the relative β-TrCP activity.

## Statistics and reproducibility

Statistical analyses were performed in MATLAB (Mathworks), and Prism (GraphPad Software). Specific statistical tests used are noted in the figure legends. No statistical method was used to determine sample size. No data were excluded from the analyses. The experiments were not randomized but usage of unbiased, automated analysis was used to ensure observer bias did not influence experimental results.

## Reporting summary

Further information on research design is available in the Nature Research Reporting Summary linked to this article.

## Data availability

The data generated in this study are provided in the Supplementary Information/Source Data File. Source data are provided with this paper.

## Code availability

All original code has been deposited at https://github.com/scappell/Cell_tracking and is publicly available.

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

## Acknowledgements

We thank Dr. Ji Luo and Stuart H. Yuspa for helpful discussions and reagent support, Dr. Yongchao Zhao for providing us the β-TrCP constructs, the Flow Cytometry Core Facility of the Center for Cancer Research at the National Cancer Institute (NCI) for technical support, and all the members of the Cappell Lab for helpful comments and support. This research was supported by the Intramural Research Programs of the National Center for Advancing Translational Sciences (NCATS; ZIA TR000455-01 to A.S.) and by the National Cancer Institute (NCI; Grant ZIA BC 011830 to S.D.C).

## Author contributions

Conceptualization, D.P. and S.D.C.; Methodology, D.P., S.C.K., J.A.C, M.M.A, G.R., A.Z., A.S, and S.D.C.; Investigation, D.P., S.C.K., J.A.C, and M.M.A.; Writing, D.P. and S.D.C; Visualization, D.P., S.C.K., J.A.C, M.M.A., and S.D.C.; Supervision, S.D.C; Funding Acquisition, S.D.C.

## Funding

## Competing interests

The authors declare no competing interests.
