## [Peer Review File · Nature Communications]

Revealing β -TrCP activity dynamics in live cells with a genetically encoded biosensorReviewers' comments:

Reviewer #1 (Remarks to the Author):

The study entitled "Revealing β -TrCP activity dynamics in live-cells with a genetically encoded biosensor" developed a fluorescent biosensor to report β -TrCP activity in single live cells. The study is interesting, and the manuscript is well written and organized. However, several major questions should be addressed carefully. Below are the comments on this manuscript.

Major issues:

1. The authors developed a fluorescent biosensor fused with human CDC25B fragment, which contains a non-canonical β -TrCP degron motif, to report β -TrCP activity. However, it is insufficient to state that "changes in the β -TrCP reporter concentration within single cells are a direct reflection of β -TrCP activity", and it can only verify that CDC25B is the substrate of SCF β -TrCP. Since no data support that the transcription and translation of this fluorescent biosensor have any fluctuation during cell cycle progression or other cellular processes, the levels of β -TrCP reporter may only reflect the levels of CDC25B, rather than β -TrCP activity during cell cycle progression. In fact, other reporters cited by the authors, such as APC/C (Ref. 34), SCFSKP2 (Ref. 27), and CRL4CDT2 (Ref. 29), have also been used as cell cycle monitors rather than E3 ligase reporters in their studies.
2. Although this manuscript is focused on protein degradation by β -TrCP, several related concepts are inappropriate. Both of β -TrCP paralogs, β -TrCP1 and β -TrCP2, with similar biochemical properties play non-redundant roles in substrate recognition as well as in the regulation of cellular processes. And cellular SCF repertoire is in a state of disequilibrium that is sustained by NEDD8 conjugation and CAND1 proteins and is modulated by substrate availability. Thus, it would be inappropriate to state that "recent evidence suggests that F-box protein themselves are under direct regulation at the level of SCF complex formation, which can influence the extent to which substrates are degraded".
3. Based on Figure 1d, this fluorescent biosensor is mainly located in the nucleus, although the authors claimed that it is located in both nucleus and cytoplasm. Several studies showed that β -TrCP1 and β -TrCP2 are mainly located in the nucleus and the cytoplasm, respectively, and β -TrCP2 plays as the dominant paralog in the regulation of several cellular processes. In addition, β -TrCP-binding motif is DpSGX1-3pS, indicating that β -TrCP can recognize DpSGXpS or DpSGXXXpS under particular conditions. Thus, this fluorescent biosensor may not fully represent the activity of β -TrCP1+2.
4. In Figure 1c, "a previous report showed that Serine 230 phosphorylation may also be involved in CDC25B degradation". However, according to Ref 32, CHK1 phosphorylates CDC25B during interphase, thus preventing the premature initiation of mitosis by negatively regulating the activity of CDC25B at the centrosome, rather than degrading it.
5. Figure 2a. There are 69 F-box proteins in mammalian cells and it would be better to detect several other F-box proteins besides FBXW7. Moreover, FBXW7 antibody (Abcam, ab109617) used in the study might be incorrect. As described in the datasheet of this antibody, FBXW7 is localized in the nucleus and the MW is below 70 kDa. However, it is well known that FBXW α , the isoform of FBXW7 localized in the nucleus, is about 100 kDa.
6. Figure 2b and 2c. Given that both β -TrCP reporter and FLAG- β -TrCP are exogenously expressed, mutant β -TrCP reporter should be included.
7. Figure 2d, 2e, and 2f. The experimental strategy was not appropriate. The ubiquitination of β -TrCP reporter binding proteins also were detected by IB with ubiquitin following IP with mVenus. Instead, the appropriate strategy is that IP with ubiquitin, followed by IB with mVenus.
8. All the endogenous Co-IP experiments should include IgG group as internal control.
9. Figure 4f. Five thousand cells seeded in a 35-mm culture dish are too many for colony formation assay. And the histogram of statistics should be included.
10. Figure 4a, and 4c, it would be important to add supplementary validation experiments confirming the specificity of β -TrCP antibody for IF.
11. Figure 5h. Given the time course of Sunitinib treatment is up to 24 hrs and the levels of many proteins alternate along with cell culture, mock treatment should be included in the same time course.
12. In Figure 5i, the increased interaction of β -TrCP with SKP1 upon treatment with sunitinib was not

remarkable, and the binding of CUL1 (neddylated or non-neddylated CUL1) with β -TrCP should be examined as well.

13. More details in the design of SKP2 reporter should be included. Importantly, no evidences support that SKP2 reporter works well.

Minor issues:

1. Figure 2b. IB of NEDD8-cullins should be included to indicate MLN4924 working well.
2. Figure 4h. It would be better to include the cleavage of PARP and caspase-3, two well-known markers of apoptosis. And the protein levels of I κ B α should be detected as well.
3. In this paper, the authors have used Actin, Tubulin, Vincullin, and HSP90 as loading control in different figures, and it is better to unify them.
4. MLN4924 is a very robust pan-SCF inhibitor. The concentration of MLN4924 used in this study, 3 μ M, was too high and may cause some unknown side effects.

Reviewer #2 (Remarks to the Author):

This paper reveals the dynamics of activity of b-TrCP through the construction of a novel and well-validated fluorescent reporter in human cells. The development of fluorescent single-cell reporters is a challenging task but usually worthwhile because the reporters often provide insight into the real-time signaling activity of individual cells. The present work is technically sound and includes all the correct controls and validation experiments to build confidence in the single-cell reporter. Enthusiasm for the work is moderate. The paper contains several interesting observations that could be helpful to those wanting to understand regulation of protein degradation through this important E3 component.

Major issues:

The primary issue with the manuscript is its ability to deliver a clear and impactful message about the role of b-TrCP in cell biology. The relatively constant level of b-TrCP activity throughout the cell cycle is an interesting result, given the observation that the expression and activity of many proteins seem to show some correlation with the cell cycle clock. This result indicates that the regulation of b-TrCP substrates lies primarily in their own phosphorylation, as the authors describe, and not in the activity of b-TrCP, which stays constant. Thus, two claims should be further developed in the paper:

1. A major theme running through the paper is the claim that "b-TrCP activity functions to maintain its substrates at discreet steady-state levels rather than completely degrading them." But the b-TrCP reporter is under the control of an artificial promoter, and – as the authors state – the levels of the reporter are a function of the rates of production (which are artificial) and degradation. So, how can the authors be sure b-TrCP is special in this regard? Is it possible that artificially controlled production of the reporter gives the false impression that "discrete" levels are maintained? Are the targets of other E3s really "completely" degraded?

2. One a related note, the engine-always-running nature of this E3 component suggest that it controls degradation at a more general level that could perhaps be tuned by upstream signaling events (i.e., RTK signaling or quiescence state). To bolster this idea, this reviewer would like to see additional b-TrCP substrates besides CDC25B and Emi1 change levels in response to upstream modulation of b-TrCP activity through treatment with sunitinib. Comparing the timing and extent of enhanced substrate degradation would build more confidence that TRK signaling modulates the activity of this potential "master" degradation regulator.

General response to all reviewers:

Thank you for providing such thoughtful feedback to our manuscript. In this revised manuscript, we have included **21 new or revised figure panels**. By incorporating your suggestions, we believe the manuscript is now much strengthened. Below you will find a point-by-point response to each of the suggestions that were raised. Your comments are shown in *italics* and our responses are shown in blue.

Reviewer #1 (Remarks to the Author):

The study entitled “Revealing β -TrCP activity dynamics in live-cells with a genetically encoded biosensor” developed a fluorescent biosensor to report β -TrCP activity in single live cells. The study is interesting, and the manuscript is well written and organized. However, several major questions should be addressed carefully. Below are the comments on this manuscript.

Major issues:

1. The authors developed a fluorescent biosensor fused with human CDC25B fragment, which contains a non-canonical β -TrCP degron motif, to report β -TrCP activity. However, it is insufficient to state that “changes in the β -TrCP reporter concentration within single cells are a direct reflection of β -TrCP activity”, and it can only verify that CDC25B is the substrate of SCF β -TrCP. Since no data support that the transcription and translation of this fluorescent biosensor have any fluctuation during cell cycle progression or other cellular processes, the levels of β -TrCP reporter may only reflect the levels of CDC25B, rather than β -TrCP activity during cell cycle progression. In fact, other reporters cited by the authors, such as APC/C (Ref. 34), SCFSKP2 (Ref. 27), and CRL4CDT2 (Ref. 29), have also been used as cell cycle monitors rather than E3 ligase reporters in their studies.

We thank the reviewer for raising this important point. As the reviewer rightly points out, a critical assumption of our reporter is that the synthesis rate of the reporter (transcription and translation) is constant throughout the cell cycle, and it was an oversight to not include data supporting that assumption. To demonstrate that the production rate of the sensor does not change throughout the cell cycle, we have now measured the sensor synthesis rate in single cells at various points in the cell cycle. Briefly, we performed live-cell imaging on hundreds of single cells asynchronously dividing. We tracked the cells to precisely identify the time since mitosis for each single cell. We then measured the change in the slope of the reporter levels following treatment with MLN-4924 in each single cell as a measure of the synthesis rate. By plotting the time since mitosis when the MLN4924 was added against the synthesis rate of the reporter in hundreds of asynchronously dividing cells we could construct an accurate view of the reporter synthesis rates throughout the entire cell cycle. We include this data in new Supplementary Fig. 1g. We find that the synthesis rate is constant throughout the cell cycle in both MCF10A and HeLa cells, consistent with the reporter being controlled by a constitutive promoter (EF1 α).

Notably, the three reporters we cited for APC/C, SCF^{Skp2}, and CRL4^{Cdt2} are all under the control of the same constitutive promoter and are therefore similarly produced at a constant rate throughout the cell cycle. The reason these reporters have been used to monitor the phase of the cell cycle in other studies is because these ubiquitin ligases are well known to only be active in specific phases of the cell cycle. For example, APC/C is activated at anaphase and is inactivated at the G1/S transition¹. Thus, the APC/C reporter only accumulates in S/G2 phase as we show in Supplementary Fig. 3a,b. Similarly, SCF^{Skp2} is activated at the G1/S transition and is inactivated at the G2/M transition. Thus, the SCF^{Skp2} reporter only accumulates in M/G0/G1 phase as we show in Supplementary Fig. 3c. Finally, CRL4^{Cdt2} is activated only when DNA replication is occurring. Thus, the CRL4^{Cdt2} reporter accumulates in G1 and G2 phases, as demonstrated by Grant et al². In other words, the reason these other ubiquitin ligase sensors have been used to monitor the cell cycle is not because the synthesis rates of these reporters fluctuate throughout the cell cycle, but rather because the activities of the ubiquitin ligases that are being monitored

by these reporters are fluctuating throughout the cell cycle. The fact that the reporters for APC/C, SCF^{Skp2}, and CRL4^{Cdt2} reflect very accurately the known temporal regulation of their respective ubiquitin ligases further supports the conclusion that the contribution of synthesis rate to the dynamics of these reporters is insignificant. Given that our β -TrCP reporter was constructed using the same design principles and is being regulated by the same constitutive promoter as these other ubiquitin ligase reporters, and that we have conducted a number of important controls, we feel confident that our reporter is similarly reflecting the activity of SCF β -TrCP.

2. Although this manuscript is focused on protein degradation by β -TrCP, several related concepts are inappropriate. Both of β -TrCP paralogs, β -TrCP1 and β -TrCP2, with similar biochemical properties play non-redundant roles in substrate recognition as well as in the regulation of cellular processes. And cellular SCF repertoire is in a state of disequilibrium that is sustained by NEDD8 conjugation and CAND1 proteins and is modulated by substrate availability. Thus, it would be inappropriate to state that “recent evidence suggests that F-box protein themselves are under direct regulation at the level of SCF complex formation, which can influence the extent to which substrates are degraded”.

We thank the reviewer for pointing this out. We have now edited the text to better reflect the complex regulation of SCF ligases by proteins such as CAND1 and NEDD8 conjugation to Cullins.

New text in introduction: “However, there is regulation of the SCF complexes as well, which can influence the extent to which substrates are degraded. For example, SCF complexes are regulated at the level of complex formation via the protein CAND1 and via NEDD8 conjugation of the Cullin subunit³⁻⁶.”

3. Based on Figure 1d, this fluorescent biosensor is mainly located in the nucleus, although the authors claimed that it is located in both nucleus and cytoplasm. Several studies showed that β -TrCP1 and β -TrCP2 are mainly located in the nucleus and the cytoplasm, respectively, and β -TrCP2 plays as the dominant paralog in the regulation of several cellular processes. In addition, β -TrCP-binding motif is DpSGX1-3pS, indicating that β -TrCP can recognize DpSGXpS or DpSGXXXpS under particular conditions. Thus, this fluorescent biosensor may not fully represent the activity of β -TrCP1+2.

The images in Fig. 1d are from an epifluorescence microscope. Given that the nucleus is thicker than the cytoplasm, the fluorescence intensity of even a non-specifically localized fluorescent protein will appear brighter in the nucleus than the cytoplasm in images taken with an epifluorescence microscope. While the reporter is indeed enriched in the nucleus relative to the cytoplasm, there is reporter localized both in the nucleus and the cytoplasm. To demonstrate this more clearly, we have performed nuclear/cytoplasmic fractionation and find the reporter is localized to both the nucleus and the cytoplasm in three cell lines tested (new Supplementary Fig. 1f).

The reviewer provides a really nice suggestion to look at the relative contribution of β -TrCP1 and β -TrCP2 to our reporter. To investigate this, we have knocked down β -TrCP1, β -TrCP2, or β -TrCP1 + β -TrCP2 in cells stably expressing the β -TrCP reporter. We found that the reporter is moderately responsive to β -TrCP1 knockdown and majorly responsive to β -TrCP2 knockdown, which supports previous observations, as the reviewer points out, that β -TrCP2 is the dominant paralog in most cells (new Supplementary Fig. 3h).

Our reporter is not tethered to one particular subcellular compartment and is therefore constantly shuttling in and out of the nucleus. Thus, if the reporter is degraded in the nucleus the reporter will re-equilibrate and we should observe a drop in reporter fluorescence in both the nucleus and in the cytoplasm. To test this, we treated HeLa cells with siRNA for β -TrCP1, β -TrCP2, or β -TrCP1 + β -TrCP2 in cells stably expressing the β -TrCP reporter. We measured the fluorescence in both the nucleus and the cytoplasm (new Supplementary Fig. 3h). As would be expected for a non-tethered reporter, we see a similar change in the dynamics of the reporter in both the nucleus and the cytoplasm. This demonstrates

that the reporter does not simply reflect β -TrCP activity from a particular sub-cellular compartment but is a whole-cell reporter of β -TrCP activity.

Additionally, we have tested the localization of β -TrCP1 and 2 in the cell lines used in this study. In HeLa cells, we found β -TrCP1 is majorly localized in the nucleus and β -TrCP2 is majorly localized in the cytoplasm. However, in MCF10A and MDA-MB-231 cells, both β -TrCP1 and β -TrCP2 are primarily localized in the nucleus (Supplementary Fig. 1f).

4. In Figure 1c, “a previous report showed that Serine 230 phosphorylation may also be involved in CDC25B degradation”. However, according to Ref 32, CHK1 phosphorylates CDC25B during interphase, thus preventing the premature initiation of mitosis by negatively regulating the activity of CDC25B at the centrosome, rather than degrading it.

We thank the reviewer for pointing this out. Given that we found the S230A mutation had no effect on the reporter dynamics, which is consistent with the reviewer’s comment and the previously published results⁷, we have removed the data and modified the text accordingly.

5. *Figure 2a. There are 69 F-box proteins in mammalian cells and it would be better to detect several other F-box proteins besides FBXW7. Moreover, FBXW7 antibody (Abcam, ab109617) used in the study might be incorrect. As described in the datasheet of this antibody, FBXW7 is localized in the nucleus and the MW is below 70 kDa. However, it is well known that FBXW α , the isoform of FBXW7 localized in the nucleus, is about 100 kDa.*

As per the reviewer’s suggestion, we have tested several other F box proteins (FBXW5, FBXO25, and FBXO31) and found no interaction with our β -TrCP sensor (new Fig. 2a).

In the NCBI database, there are 3 isoforms of FBXW7 that are comprised of 707, 627, and 589 amino acids. The predicted translated molecular weights of these isoforms are 79, 70, and 67 kDa respectively. It is likely our cells primarily express isoform 2, hence the 70kDa band we observe. To validate that our antibody is detecting FBXW7, we used siRNA to knockdown FBXW7 and see a decrease in a band at the molecular weight corresponding to FBXW7 isoform 2. Additionally, we confirmed our siRNA worked because we observed an accumulation of FBXW7’s known substrate cMyc (Supplementary Fig. 3f).

6. *Figure 2b and 2c. Given that both β -TrCP reporter and FLAG- β -TrCP are exogenously expressed, mutant β -TrCP reporter should be included.*

As per the reviewer’s suggestion, we have now included the mutant β -TrCP reporter in these figures (see new Fig. 2b,c). The mutant β -TrCP reporter is not affected by overexpression of FLAG- β -TrCP.

7. *Figure 2d, 2e, and 2f. The experimental strategy was not appropriate. The ubiquitination of β -TrCP reporter binding proteins also were detected by IB with ubiquitin following IP with mVenus. Instead, the appropriate strategy is that IP with ubiquitin, followed by IB with mVenus.*

As per the reviewer’s suggestion, we have modified the experimental design and performed a ubiquitin pulldown and mVenus detection by IB. We have used both endogenous Ub and His-Ub mediated pulldown and found similar results (new Fig. 2d-f and new Supplementary Fig. 2d). We confirmed the β -TrCP reporter is ubiquitinated but the mutant β -TrCP reporter is not (Fig. 2d and Supplementary Fig. 2d). We observed an increase in the ubiquitination of the reporter when we over-expressed FLAG- β -TrCP1 but not Δ FLAG- β -TrCP1 (Fig. 2e). Finally, the ubiquitination of the reporter is diminished after knocking down β -TrCP1,2 or Cullin1 (Fig. 2f).

8. *All the endogenous Co-IP experiments should include IgG group as internal control.*

We have included IgG control for each condition in the IP presented in the revised version.

9. *Figure 4f. Five thousand cells seeded in a 35-mm culture dish are too many for colony formation assay. And the histogram of statistics should be included.*

For this experiment, we used 35mm dishes and initially plated 2500, 5000, and 10,000 cells. We found that for this cell line, under our growth conditions, that 5000 cells yielded the best results. We have now included a bar graph with results from 3 biological repeats as well as the statistics showing a significant difference in colony formation (new Supplementary Fig. 5k).

10. *Figure 4a, and 4c, it would be important to add supplementary validation experiments confirming the specificity of β -TrCP antibody for IF.*

We thank the reviewer for raising this point. As per the reviewer's suggestion, we have included the validation of the β -TrCP antibody by IF, shown in new Supplementary Fig. 5e. We include both images as well as a single-cell quantification of the fluorescence intensity of both control and β -TrCP1,2 siRNA treated cells.

11. *Figure 5h. Given the time course of Sunitinib treatment is up to 24 hrs and the levels of many proteins alternate along with cell culture, mock treatment should be included in the same time course.*

We thank the reviewer for raising this point. We have now included a time course of DMSO treatment. Additionally, we have checked several other β -TrCP substrates including Dyrk1a, I κ B α , and PFKB3 alongside Emi1, endogenous Cdc25B, and the mVenus-tagged reporter to strengthen our conclusion that the RTK signaling pathway controls β -TrCP activity, shown in new Fig. 5h.

12. *In Figure 5i, the increased interaction of β -TrCP with SKP1 upon treatment with sunitinib was not remarkable, and the binding of CUL1 (neddylated or non-neddylated CUL1) with β -TrCP should be examined as well.*

As per the reviewer's suggestion, we have included Cullin1 in the IP and quantified the respective binding to β -TrCP in each condition (Fig. 5i).

13. *More details in the design of SKP2 reporter should be included. Importantly, no evidences support that SKP2 reporter works well.*

We thank the reviewer for pointing out this oversight. The Skp2 reporter has been tested and validated in many previous studies² as it is widely used as one of the original FUCCI reporters⁸. We did not initially include a description of this sensor in the manuscript given its wide availability and usage in the field. However, given the importance of this reporter to the interpretation of our results, the reviewer is correct in pointing out we should include these details in the manuscript. We have now edited the text and methods section to include more details about this reporter. Briefly here, it is comprised of a fragment of the SCF-Skp2 substrate Cdt1 (aa30-120) fused to mVenus and contains a constitutive phospho-mimetic linker which ensures that the reporter reflects SCF-Skp2 activity². In addition to providing citations for the construction and validation of this reporter, we include some of our own validation data in Supplementary Fig. 6b demonstrating the reporter increases following MLN-4924 treatment and decreases following treatment with Cycloheximide.

Minor issues:

1. Figure 2b. IB of NEDD8-cullins should be included to indicate MLN4924 working well.

We have now included NEDD8 immunoblotting in Fig. 2b. We observe a loss in a high-molecular weight band corresponding to NEDD8-Cullin upon treatment with MLN-4924, confirming that the inhibitor is working well.

2. Figure 4h. It would be better to include the cleavage of PARP and caspase-3, two well-known markers of apoptosis. And the protein levels of I κ B α should be detected as well.

As per the reviewer's suggestion, we have now included protein levels of I κ B α (Fig. 4h) and cleaved PARP1 and caspase 3 shown in new Supplementary Fig. 5m.

3. In this paper, the authors have used Actin, Tubulin, Vincullin, and HSP90 as loading control in different figures, and it is better to unify them.

Thank you for this suggestion. In an effort to better unify our loading controls, we have only used vinculin as the loading control for all the revised experiments included in this version of the manuscript.

4. MLN4924 is a very robust pan-SCF inhibitor. The concentration of MLN4924 used in this study, 3 μ M, was too high and may cause some unknown side effects.

We thank the reviewer for raising this point. 3 μ M is indeed above the IC₅₀ of the drug. We chose this dose to ensure we have maximally inhibited cullin neddylation in our cells. However, to verify that degradation of our β -TrCP reporter can be rescued at lower doses of MLN-4924, we conducted a new experiment by treating cells with increasing concentrations of MLN-4924 (500nM-3 μ M) and blotting for our mVenus-tagged reporter. We see at even the lowest dose of MLN-4924 (500nM) that the reporter is stabilized (new Supplementary Fig. 2c). Therefore it is unlikely that unknown side-effects of MLN-4924 are responsible for the dynamics of the β -TrCP reporter we observed.

Reviewer #2 (Remarks to the Author):

This paper reveals the dynamics of activity of β -TrCP through the construction of a novel and well-validated fluorescent reporter in human cells. The development of fluorescent single-cell reporters is a challenging task but usually worthwhile because the reporters often provide insight into the real-time signaling activity of individual cells. The present work is technically sound and includes all the correct controls and validation experiments to build confidence in the single-cell reporter. Enthusiasm for the work is moderate. The paper contains several interesting observations that could be helpful to those wanting to understand regulation of protein degradation through this important E3 component.

Major issues:

The primary issue with the manuscript is its ability to deliver a clear and impactful message about the role of β -TrCP in cell biology. The relatively constant level of β -TrCP activity throughout the cell cycle is an interesting result, given the observation that the expression and activity of many proteins seem to show some correlation with the cell cycle clock. This result indicates that the regulation of β -TrCP substrates lies primarily in their own phosphorylation, as the authors describe, and not in the activity of β -TrCP, which stays constant. Thus, two claims should be further developed in the paper:

1. A major theme running through the paper is the claim that " β -TrCP activity functions to maintain its

substrates at discreet steady-state levels rather than completely degrading them.” But the β -TrCP reporter is under the control of an artificial promoter, and – as the authors state – the levels of the reporter are a function of the rates of production (which are artificial) and degradation. So, how can the authors be sure β -TrCP is special in this regard? Is it possible that artificially controlled production of the reporter gives the false impression that “discrete” levels are maintained? Are the targets of other E3s really “completely” degraded?

We thank the reviewer for this helpful suggestion. To demonstrate that the production rate of the sensor does not change throughout the cell cycle, we have measured the sensor synthesis rate in single cells at various points in the cell cycle. We have included this data in new Supplementary Fig. 1g. We find that the synthesis rate (a combination of transcription and translation) is constant throughout the cell cycle in both MCF10A and HeLa cells. Thus, we can conclude that the changes in the reporter levels are not due to changes in synthesis rate, but rather solely due to changes in β -TrCP activity.

To demonstrate that β -TrCP is unique in its role to maintain substrates at discreet steady-state levels rather than to degrade them completely, we have compared our β -TrCP sensor to two similar E3 ubiquitin ligase sensors for APC/C and SCF^{Skp2}, also well known as the original FUCCI reporters⁸. Notably, these two sensors are under the control of the same constitutive promoter (EF1 α) as our β -TrCP reporter. This allows for a much more direct comparison of the various ubiquitin ligase sensors. For the APC/C sensor, we find that it is completely degraded down to background levels at anaphase (see Supplementary Fig. 3a,b), and the reporter remains completely degraded until the start of S-phase when the APC/C is known to inactivate¹. We have previously demonstrated that the APC/C reporter levels are “completely” degraded by comparing the reporter levels to background fluorescence in Cappell et al (see Supplemental Figure S2A)¹. For the SCF^{Skp2} sensor, we find that it is completely degraded down to background levels during S phase and the reporter levels remain completely degraded until the end of G2 phase, consistent with when Skp2 levels are known to be present during the cell cycle (see Supplementary Fig. 3c). A similar ubiquitin ligase sensor for CRL4^{Cdt2} was previously published² and the authors similarly found that its levels were completely degraded during S phase, when CRL4^{Cdt2} is known to be active. Thus, for 4 separate ubiquitin ligase sensors all under the control of the same constitutive promoter, β -TrCP is unique in that it fails to fully degrade the reporter. We have updated the text to better report these observations.

2. One a related note, the engine-always-running nature of this E3 component suggest that it controls degradation at a more general level that could perhaps be tuned by upstream signaling events (i.e., RTK signaling or quiescence state). To bolster this idea, this reviewer would like to see additional β -TrCP substrates besides CDC25B and Emi1 change levels in response to upstream modulation of β -TrCP activity through treatment with sunitinib. Comparing the timing and extent of enhanced substrate degradation would build more confidence that TRK signaling modulates the activity of this potential “master” degradation regulator.

We thank the reviewer for raising this point. We have now checked multiple other β -TrCP substrates including Dyrk1a, Ikb α , and PFKB3 (new Fig. 5h). We find that similar to CDC25B and Emi1, these other β -TrCP substrates are controlled by RTK signaling, supporting our conclusion that RTK signaling functions as a “master” degradation regulator of β -TrCP substrates.

References:

- 1 Cappell, S. D., Chung, M., Jaimovich, A., Spencer, S. L. & Meyer, T. Irreversible APC(Cdh1) Inactivation Underlies the Point of No Return for Cell-Cycle Entry. *Cell* **166**, 167-180, doi:10.1016/j.cell.2016.05.077 (2016).

- 2 Grant, G. D., Kedziora, K. M., Limas, J. C., Cook, J. G. & Purvis, J. E. Accurate delineation of cell cycle phase transitions in living cells with PIP-FUCCI. *Cell cycle* **17**, 2496-2516, doi:10.1080/15384101.2018.1547001 (2018).
- 3 Reitsma, J. M. *et al.* Composition and Regulation of the Cellular Repertoire of SCF Ubiquitin Ligases. *Cell* **171**, 1326-1339 e1314, doi:10.1016/j.cell.2017.10.016 (2017).
- 4 Saha, A. & Deshaies, R. J. Multimodal activation of the ubiquitin ligase SCF by Nedd8 conjugation. *Mol Cell* **32**, 21-31, doi:10.1016/j.molcel.2008.08.021 (2008).
- 5 Goldenberg, S. J. *et al.* Structure of the Cand1-Cul1-Roc1 complex reveals regulatory mechanisms for the assembly of the multisubunit cullin-dependent ubiquitin ligases. *Cell* **119**, 517-528, doi:10.1016/j.cell.2004.10.019 (2004).
- 6 Pierce, N. W. *et al.* Cand1 promotes assembly of new SCF complexes through dynamic exchange of F box proteins. *Cell* **153**, 206-215, doi:10.1016/j.cell.2013.02.024 (2013).
- 7 Schmitt, E. *et al.* CHK1 phosphorylates CDC25B during the cell cycle in the absence of DNA damage. *J Cell Sci* **119**, 4269-4275, doi:10.1242/jcs.03200 (2006).
- 8 Sakaue-Sawano, A. *et al.* Visualizing spatiotemporal dynamics of multicellular cell-cycle progression. *Cell* **132**, 487-498, doi:10.1016/j.cell.2007.12.033 (2008).

REVIEWER COMMENTS

Reviewer #1 (Remarks to the Author):

The authors have addressed most of my concerns. However, in supplementary Fig. 1f, GAPDH can still be detected in the cytosolic fraction, indicating that nuclear fraction contains contamination of cytosolic proteins. In addition, the authors should indicate which band is beta-TrCP2, since two bands were detected for beta-TrCP2. Given that the location of beta-TrCP2 is inconsistent with the previous studies, the specificity of this antibody need to be tested using siRNA-based knockdown.

Reviewer #2 (Remarks to the Author):

I am satisfied with the authors' responses and new data supporting my two critiques and now recommend the paper for publication.

General response to all reviewers:

In this revised manuscript, we have included **1 revised figure panel**. Below you will find a point-by-point response to each of the suggestions that were raised. Your comments are shown in *italics* and our responses are shown in **blue**.

Reviewer #1 (Remarks to the Author):

The authors have addressed most of my concerns. However, in supplementary Fig. 1f, GAPDH can still be detected in the cytosolic fraction, indicating that nuclear fraction contains contamination of cytosolic proteins. In addition, the authors should indicate which band is beta-TrCP2, since two bands were detected for beta-TrCP2. Given that the location of beta-TrCP2 is inconsistent with the previous studies, the specificity of this antibody to be tested using siRNA-based knockdown.

As the reviewer noted, there is a small contamination of cytosolic proteins in the nuclear fraction in Supplementary Fig. 1f (for the HeLa and MDA-MB-231 cell lysates). We have repeated these fractionation experiments to yield cleaner results (see new Supplementary Fig. 1f). Asynchronously cycling cells were harvested followed by fractionation and immunoblotting. With the fresh samples, we did not have contamination of cytosolic proteins in the nucleus, and we did not observe the second, non-specific band with FBXW11 antibody, likely due to cleaner fractionation, fresh samples, and freshly prepared antibody. Furthermore, we have now included arrows to better identify the bands corresponding to β -TrCP1 and β -TrCP2. Finally, as suggested by the Reviewer, we have also run MDA-MB-231 cell lysates transfected with siRNA targeting β -TrCP1 and β -TrCP2 to better identify the relevant bands and their localization.

Reviewer #2 (Remarks to the Author):

I am satisfied with the authors' responses and new data supporting my two critiques and now recommend the paper for publication.

We thank the reviewer for taking the time to provide helpful feedback.